# Evaluation of Forest Industry Scenarios to Increase Sustainable Forest Mobilization in Regions of Low Biomass Demand

**Fernando Pérez-Rodríguez** [1],* and **João C. Azevedo** [2]

1 Föra Forest Technologies Sll, Campus Duques de Soria s/n, 42004 Soria, Spain
2 Centro de Investigação de Montanha, Instituto Politécnico de Bragança, Campus de Santa Apolónia, 5300-253 Bragança, Portugal; jazevedo@ipb.pt
* Correspondence: fernando.perez@fora.es; Tel.: +34-627-060-844

**Abstract:** There is an increasing interest in forest biomass for energy throughout Europe, which is seen as a way of promoting forest mobilization and economic development locally, in particular in regions where forest biomass is available but its use is limited by lack of demand. This study was conducted to define, evaluate and select viable forest industry scenarios to increasing forest mobilisation in the North of Portugal using AppTitude®, a Forest Management Decision Support Systems (FMDSS) considering spatially explicitly supply (biomass growth and yield), demand (industry), and supply–demand interactions (markets). The protocol followed combined a set of indicators of sustainable forest management to guide the selection of the best industry solutions in terms of location, dimension, forest biomass and other variables defined as objectives. The simulations allowed the selection of a small set of industry scenarios compatible with an existing plant outside the study area, increasing wood mobilization, preventing overexploitation and competition among industries but increasing value and price of forest biomass. The results of the application of this FMDSS showed that introducing new biomass plants in the region will increase sustainable forest mobilization and related local development. AppTitude® revealed to be a powerful and reliable tool to assist forest planning.

**Keywords:** heuristics; stochastic; AppTitude®; linear programming; MAUT

## 1. Introduction

The demand for forest biomass for energy has been increasing throughout Europe [1,2] and it is expected to grow further in the near future [3]. At the beginning of the millennium, expectations had already been of great demand in the coming decades (e.g., [4]). Today, biomass is gaining importance within the overall goal of the contribution of renewable energies to the composite pool of energy sources [5] in the context of multifunctional forestry [6]. Biomass energy industry has shown a tremendous increase throughout Europe in recent years, growing, for example, by 97.6% overall between 2009 and 2013 [1,7]. The general trend of growth of forest biomass supply and demand, however, is not homogeneous across Europe [2,8]. Countries such as Sweden or Germany are net biomass for energy exporters while countries such as the Netherlands or Denmark are net importers [5]. Similar patterns can be observed at country or region scales. For example, although Portugal has been one of the major wood pellets producing and exporting countries in Europe [5,9], most of the production takes place in the north and central coastal regions closer to the coast [10]. In the interior areas of the country, the demand of wood for energy has suffered little influence from the overall fast-growing national and international wood energy markets and the use of wood for energy follows traditional local patterns [11].

The increasing demand of forest biomass for energy is often seen as an opportunity for a bio-based economy [12], potentially triggering wood mobilisation in general and, in particular, in regions and forest types for which there traditionally has not been demand, ultimately contributing to rural development in these regions [13,14]. The local establishment of industrial energy units using forest biomass (power or pellets production plants) is, therefore, a possible outcome of this increasing demand in areas currently not covered by forest industry, but where forest biomass is abundant and apparently available for mobilisation. In a fire-prone country like Portugal, demand from the biomass energy sector is additionally seen as a way of contributing to the reduction in fire hazard by decreasing the horizontal and vertical continuity of fuel (biomass) in forests and landscapes [15].

The introduction of new forest industries in a region should be supported by sustainability assessments addressing economic, social and environmental aspects [16] such as: (i) the economic viability and risk of the investment, (ii) the environmental resilience and the sustainability of the resource supply, and (iii) the social impact of the investment and the potential conflicts with other biomass uses or industries. These assessments should start by considering the actual supply cycle/chain and market of the resource/product [17], envisioning possibilities for changes in the region aiming to increase investments in transforming industries (pellets, bioenergy, etc.).

The value of forest biomass in general holds a large subjective component [18], causing it to be vulnerable to fluctuations according to supply and demand [19]. Moreover, forests, as complex and dynamic systems, offer a diversity of other goods and services generating different types of perceptions and expectations from stakeholders that can create a series of interactions among them. Forest Management Decision Support Systems (FMDSS) have been gaining importance in recent decades under this framework, producing methods and tools to address these complex systems and help to take decisions [20]. In general, an FMDSS is a hierarchy or structure of methods and tools whose main goal is to look for a solution for a problem in forestry [21]. An FMDSS encompasses growth and yield modelling [22,23], spatial analysis [24] and interactions among stakeholders [25], considering different temporal and spatial scales and choices and incorporating hazard, risk and uncertainly analysis [26,27]. Finding a solution for a complex problem requires large calculation capabilities. The rapid evolution of computer processing power has made it possible to apply complex FMDSS to real-world problem solving (e.g., Eureka in Sweden [28], SADfLOR [29] and FlorNExT Pro© [30] in Portugal) whose development is a challenge for future decision making.

The present study aimed to apply the FMDSS AppTitude®, developed and established in the Nordeste region (North of Portugal) to support increasing forest mobilization, in the definition and evaluation of scenarios representing the implementation of forest industry plants locally. The framework used for the development of AppTitude® and its application in this study addressed supply, demand, and supply–demand interactions according to forest growth and yield dynamics, demand from industry, and conflicts among industries for the use of wood established on a spatially explicit basis. The objectives of this study were to (i) define forest industry scenarios with the potential to increase forest mobilization in the Nordeste region in Portugal, (ii) assess the impacts of these scenarios on forest resources and on forest mobilisation, and (iii) select the most viable and sustainable scenarios for the region.

## 2. Materials and Methods

### 2.1. Study Area

This study was conducted in the Nordeste Transmontano region, Northeastern Portugal, (Figure 1), hereafter Nordeste encompassing a total land area of 527,705 há and eight municipalities (Afândega da Fé, Bragança, Macedo dos Cavaleiros, Miranda do Douro, Mirandela, Mogadouro, Vimioso and Vinhais). The region presents low levels of wood mobilization due to a combination of factors including lack of information, lack of awareness regarding forest potential, and lack of investment in forestry and the forest-based industry. However, forests in this region are relatively

abundant (25% of the land cover, overall) offering currently a non-neglectable yield level. Maritime pine (*Pinus pinaster*) is the major forest species in area with more than 28,000 ha [31].

The potential establishment of forest industries (sawmills, biomass-fired power plants, or wood pellets producing plants) are seen locally as a way of valorising forest resources and increasing forest management intensity and wood mobilization, contributing to the sustainable development of the region as well as to decreasing fire hazards [32]. In this study, we established hypothetical industry units, in addition to an existing industry already using resources in the region, as possibilities for increasing wood demand regionally, which were tested with the FMDSS developed for Nordeste.

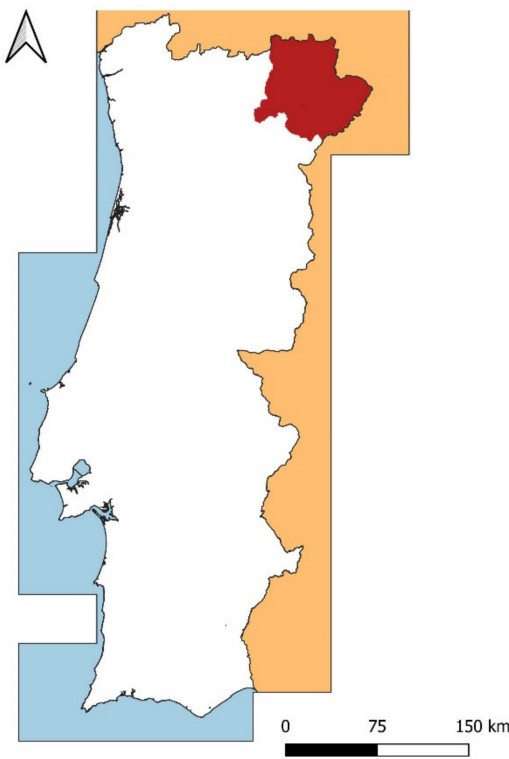

**Figure 1.** Location of the study area, the Nordeste region (in red), in Portugal (in white).

## 2.2. Approach and Model Development

Heuristics, "a strategy that ignores part of the information with the goal of making decisions more quickly, frugally, and/or accurately than more complex methods" [33], provides solutions to complex problems that cannot be solved in any other way, for example by structuring these problems into simple divisions of parts with a defined goal [34]. In this study, the problem consisted of assessing spatial and temporal interactions of actors (forest industries) with resources (forest biomass) in a defined area (Nordeste, Portugal). We assumed this complex system to be made up of three components:

(i)     Supply, or how much and where the resource (forest biomass) is available in the region;
(ii)    Demand, or what characteristics of the resource are of interest to the users (forest industry);
(iii)   The interactions between supply and demand in the study area.

To address this problem, we used AppTitude®, a FMDSS tool developed with the aim of supporting forest mobilization in the Nordeste region, that applies the rationale and the methodological framework followed in this study [35] (Figure 2).

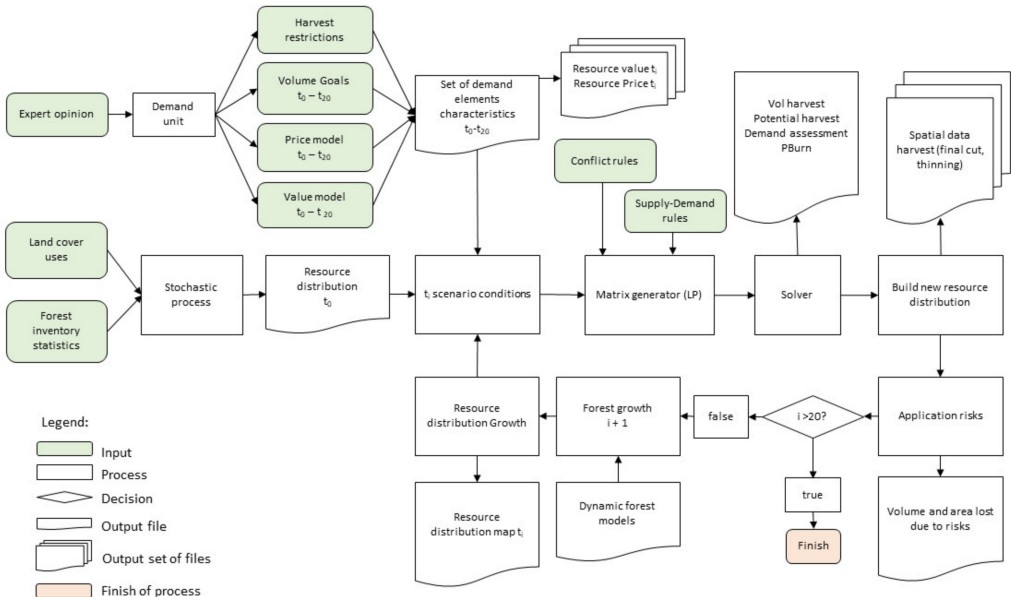

**Figure 2.** Flow diagram of the simulation procedure applied in for supply, demand, and supply–demand interactions assessment. AppTitude® implements quantity, value and price models to define scenario conditions required for the linear programming matrix that is iterated and solved annually for a 20-y period applying forest growth models in the area of study.

AppTitude® combines a set of methods in a complex decision support system applied in a 5-step protocol to define scenarios, inputs, and restrictions and to reach goals, targets and objectives in order to find a satisfactory solution for a given problem (Figure 3). It was used in this study to address the effects of forest industries on forest resources and the interactions among competing industries and to select the best industry scenarios for the region. The procedures for ranking and selecting solutions are based on a series of objectives (Table 1). These particular objectives are of two types: necessary and priority. Necessary objectives are mandatory and take into consideration the sustainability of activities and resilience of the region to absorb management operations of all industries in the region (Objective 1), from the point of view of supply, or resource availability and the of operations of each industry throughout the simulation period (Objective 2), from the point of view of demand. The priority objectives are not mandatory but are used to rank the results of the model to select the best scenarios. We considered in this study as priority objectives, from the point of view of demand, to maximize volume harvest (Objective 3) and to maximize valorisation of maritime pine in the region (Objective 4) and, from the point of view of the supply–demand iteration, to maximize the managed area (Objective 5), the lowest competition among industries (Objective 6) and to decrease forest fires (Objective 7). These objectives were monitored and verified through a series of indicators (Id) that will be described in the following sections.

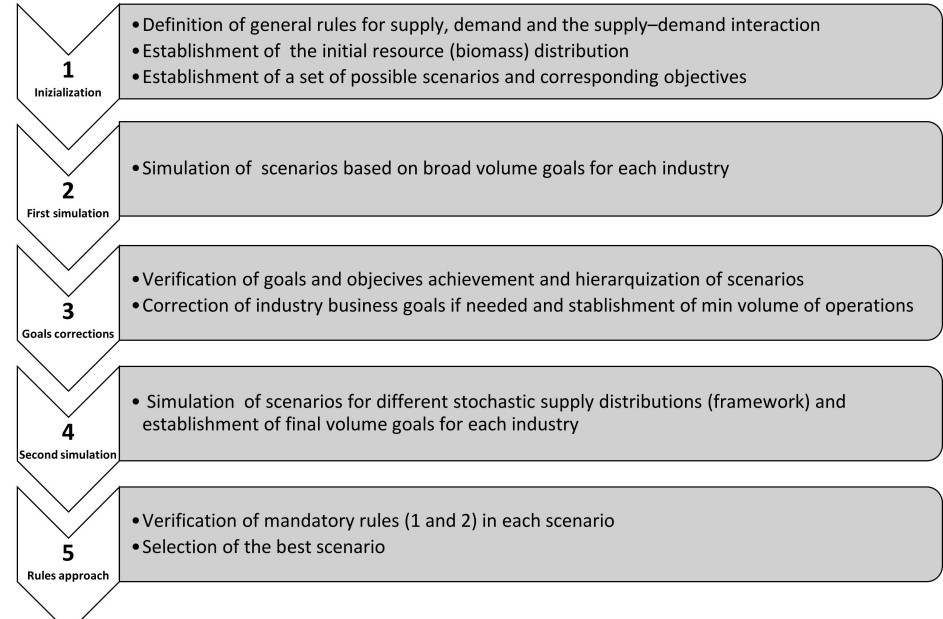

**Figure 3.** Protocol for the application of the AppTitude® process/methodology.

**Table 1.** Objectives established for the ranking and selection of solutions for the particular case of this research (Nordeste region, Portugal).

| Objective | Description | Component Directed to | Type |
|:---:|:---:|:---:|:---:|
| 1 | Assure resilience of the region to absorb the operations of all industries | Supply | Necessary |
| 2 | Assure continuity in the operations of each industry throughout the simulation period | Demand | Necessary |
| 3 | Maximize volume harvest with respect to control scenario | Demand | Priority |
| 4 | Maximize resource valorisation with respect to control scenario | Demand | Priority |
| 5 | Maximize managed area in the study region | Supply–demand Interaction | Priority |
| 6 | Minimize competition among industries | Supply–demand Interaction | Priority |
| 7 | Minimize effects of forest fires | Supply–demand Interaction | Priority |

*2.3. Supply Assessment*

2.3.1. Quantity Model

The first step in structuring a supply–demand problem is to understand the availability of the resource [36]. The supply assessment for the Nordeste was based on both the spatial distribution of forest biomass in the study area (spatial dimension) and changes in forest biomass over time in every location (temporal dimension). The biomass distribution was done based on the Land Use/Land Cover assessment COS2007 [31] data and through stochastic modelling based on statistical distributions generated from the National Forest Inventory [37]. This approach allows the integration of modelling tools of several families to accurately describe the spatial and temporal dynamics of forest resources related to forest stand growth and forest management operations such as thinning and felling. It can also to overcome a common problem in forest data sources, that is, the lack of detailed data on the spatial distribution of forest stand attributes. To build the stochastic model, we disaggregated forest

spatial data into n rows by m columns 1-ha resolution units. For each spatial unit $U_{ij}$ at the ith row and jth column locations of the $M_{nm}$ matrix, we assigned the variables Species (SP), Stand Age (t), Site Index (SI), Dominant Height ($H_0$), Density (N), Density factor (FN), Basal Area (G), Number of previous thinnings (NT), and Age of the stand when it was last thinned (AT), based on data from the aforementioned Land Use/Land Cover database and from initialization processes based on statistical distributions established from the Portuguese National Forest Inventory [37] plot data. Initialization consisted of: (i) establishing the age of each spatial unit as a pseudorandom variable using the age probability distribution obtained from the National Forest Inventory data; (ii) initializing $H_0$ using the SI spatial distribution established using National Forest Inventory plot data; (iii) initializing G and N using the equations of [38,39], respectively; (iv) assigning FN to each spatial unit using Land Use/Land Cover data [31].

Forest biomass available in the landscape was a function of the growth of stands and management operations applied in each spatial unit over time based on growth and yield models, projecting forest biomass at both the stand and landscape levels. For our area of study, we used previously developed dynamic models for maritime pine [38–40], after local validation [41]. The application of silvicultural practices in each spatial unit according to the supply and demand conditions simulated (no management, thinning, and felling) was based on the forest management tool FlorNExT® [41]. Moreover, we introduced a fire risk factor in the model combining an empirical probability of fire occurrence in case of ignition [42] with the ignition probability model of [43], resulting in a probability of resource loss in each $U_{ij}$.

Considering the infinite number of possible spatial distributions of forest resources according to the stochastic distribution models used, we tested six possible baseline distributions for the initialization of the model (Appendix A). One of these distributions, Forest Stochastic Distribution 0 (FSDo), was used in the first simulation of the model (step 1 in Figure 3) with all scenarios (see Section 2.4.1). The remaining distributions (FSD1 to FSD5) were used in the second simulation (step 4 in Figure 3) for the scenarios selected in the first simulation to address the effects of variability in forest data on the indicators used to describe the behaviour of the model.

Two indicators were used to assess trends of supply and, in particular, to verify that objective 1 was met (Resilience of the region to absorb the operations of all industries):

**Id 1**: Overall change in volume in the simulation period ($m^3$/yr): Slope of the linear regression fit to the volume in the region over the 20-y simulation period. When positive, it indicates that the supply to industries in the entire simulation period does not decrease volume in stands in the region in this period;

**Id 2**: Overall change in volume in the last five simulation years ($m^3$/y): same as above for the last five years of the simulation.

2.3.2. General Model Restrictions

The extremely large number of spatial units in the study area and the high number of combinations of all possible solutions for these units, make the number of variables involved in the linear programming problem (supply–demand component) very high. To reduce the complexity of the problem, we applied the following restrictions in the supply component of the model:

(i)　Forest spatial units are even-aged maritime pine stands;
(ii)　Species composition in a particular $U_{ij}$ is constant in time;
(iii)　Site Index (SI) for a particular $U_{ij}$ is constant in time;
(iv)　The Density Factor (FN) for a particular $U_{ij}$ is constant in time;
(v)　When a unit is harvested, Density is reset to 2000 trees/ha the following year.
(vi)　When thinning is applied, no other operation (either thinning or felling) can be applied again in less than 6 years.
(vii)　No spatial restrictions apply to thinning or felling.

## 2.4. Demand Assessment

### 2.4.1. Demand Scenarios

Our analysis is based on the test of 10 different demand scenarios built combining three forest-related industry locations (I, II, and III), four industrial activities (B1, B2, B3, S1) (Figure 4) and a diversity of supply–demand relationships (Table 2):

- **Biomass I (B1)**: an existing pellets production plant located just west the study area (Chaves);
- **Biomass II (B2)** and **Sawmill (S1)**: a plant with two divisions, a biomass-fired power plant and a sawmill, located in Bragança, the largest city in the region;
- **Biomass III (B3)**: a biomass-fired power plant located in Vimioso, in the east of the region, where forests, although young, are relatively abundant.

These scenarios address specific thinning/harvest conditions and ranges of prices for each industry (Appendix B, Tables A2 and A3, respectively). The evaluation of these 10 scenarios was conducted through the verification of objectives in Table 1 using the following viability indicators for individual industries. Objective 2 and 3 were assessed with indicators 3 and 4, respectively:

**Id 3**: Continuous volume demand in every year of the simulation period (Y/N): objective 2 is met when there is demand in every year of the period simulated;

**Id 4**: Total volume demand (m3): Summation of all volume harvest of all industry in each scenario.

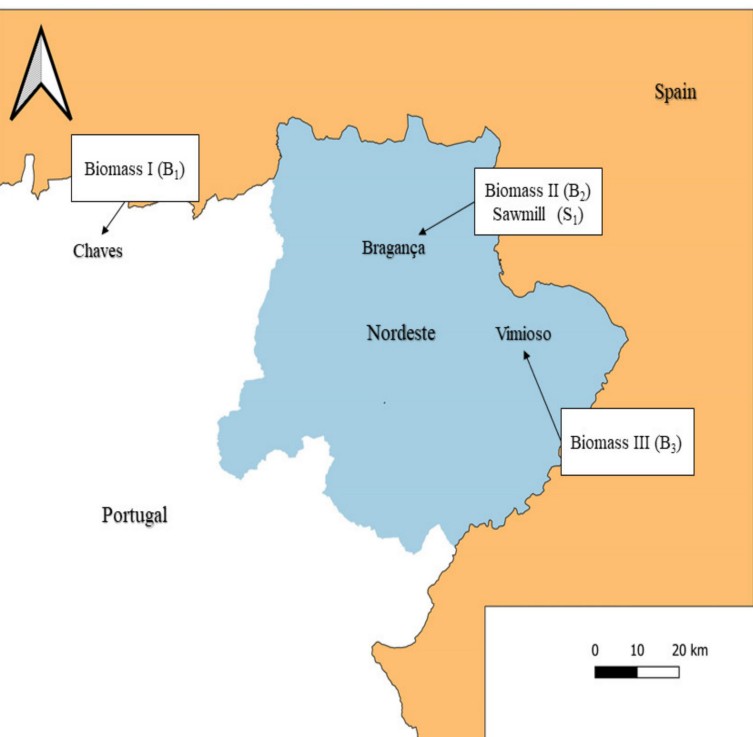

**Figure 4.** Location of forest industries considered in the 10 demand scenarios tested in the Nordeste region, Portugal.

**Table 2.** Description of scenarios tested in the Nordeste region, Portugal.

| Name | Description |
|---|---|
| Scenario 1 (Control) | Corresponds to Biomass I (B1), where demand comes from an already existing pellets production plant located outside (Chaves) but consuming biomass from the study area; all the available pine biomass in the region can be used by this plant only; this is the control scenario and it will be used as reference for other scenarios |
| Scenario 2 | Scenario 1 + Biomass II (B2), a bioenergy plant located in Bragança; no supply restrictions |
| Scenario 3 | Scenario 2 + Biomass III (B3), a bioenergy plant located in Vimioso; no supply restrictions |
| Scenario 4 | Scenario 1 + Biomass III (B3); no supply restrictions |
| Scenario 5 | Scenario 1 + Biomass II (B2) and Sawmill (S1); supply in B1 and B2 restricted to biomass extracted by thinning |
| Scenario 6 | Scenario 5; biomass supply to B1 and B2 from thinning and felling when average dbh < 25 cm at stand age 30 years |
| Scenario 7 | Scenario 5 + Biomass III (B3); supply in B1, B2 and B3 from thinning and felling when average dbh < 25 cm at stand age 30 years |
| Scenario 8 | Scenario 7 + Sawmill (S1); supply in B1 limited to thinning and felling in stands of average dbh < 25 cm) |
| Scenario 9 | Scenario 1 + Sawmill (S1); supply in B1 limited to thinning and felling in stands of average dbh < 25 cm) |
| Scenario 10 | Scenario9 + Biomass II (B2); in B2 maximum annual volume of 5000 m$^3$ |

dbh: diameter at the breast height (1.3 m above ground).

### 2.4.2. Value and Price Models

We established the demand component of the model based on biomass valuing and pricing. For value, we adopted the definitions according to the value "whatever I want in a product", since it can be converted in terms of subjective satisfaction concerning products [18]. Different stakeholders look for different characteristics in a product, which makes the value of that product dependent on the point of view of different groups of interest. A forest stand, for example, may not have the same value for the forest industry, tourists or mushroom pickers. The question then becomes: what is each stakeholder looking for and how can that information be converted into value? We addressed these questions with the Analytic Hierarchy Process (AHP) [44] and Multi-Attribute Utility Theory (MAUT) [45] multi-criteria analysis methods. AHP is one of the most widely used methods around the world and it has been applied to decision making in many fields [46], including forestry (e.g., [47,48]. AHP is based on the pairwise comparison of criteria in a decision scheme as well as on the alternatives under each criterion using a certainty scale of priorities [49]. MAUT is similar to AHP in many ways [50], but instead of pairwise criteria comparisons, it uses utility functions ranging from 0 to 10 for the level of satisfaction for each criterion [51]. There are several examples of applications of MAUT in forestry, for example [52,53].

AHP was used in our framework to obtain weights for criteria in a decision tree and MAUT to link weights with the weight utility model for each criterion, or data category, of spatial datasets associated to each criterion. The relationship between criteria and spatial information was synthetized using a utility function where attributes were given weights according to priorities (Table 3). The value model considered published information [54–57] for the establishment of criteria and weights assigned to criteria/sub-criteria (Figure 5). Sub-criteria road transport $CO_2$ emissions and road transport distance were considered within policy criteria, mechanizable (slope class) and road transport fuel consumption within economy, and protection areas, land use, and distance to rivers within environment. Weights, defined for the present year ($t_0$) and for simulation year 20 ($t_{20}$) (Figure 6), were constant

across industries. Changes in value over time were based on linear relationships of criteria in the decision model. The output of the value model was a spatial distribution of weights of the resource from the perspective of the forest biomass industry.

**Table 3.** Utility function for the establishment of weights for criteria.

| Score | Description |
|---|---|
| Null | The criterion is not important, or it is not evaluated (out of the function) |
| 0 | The criterion is not available for the tree hierarchy (for technical or legal limitations) |
| 1 to 10 | 1: minimum weight; 10: maximum weight |

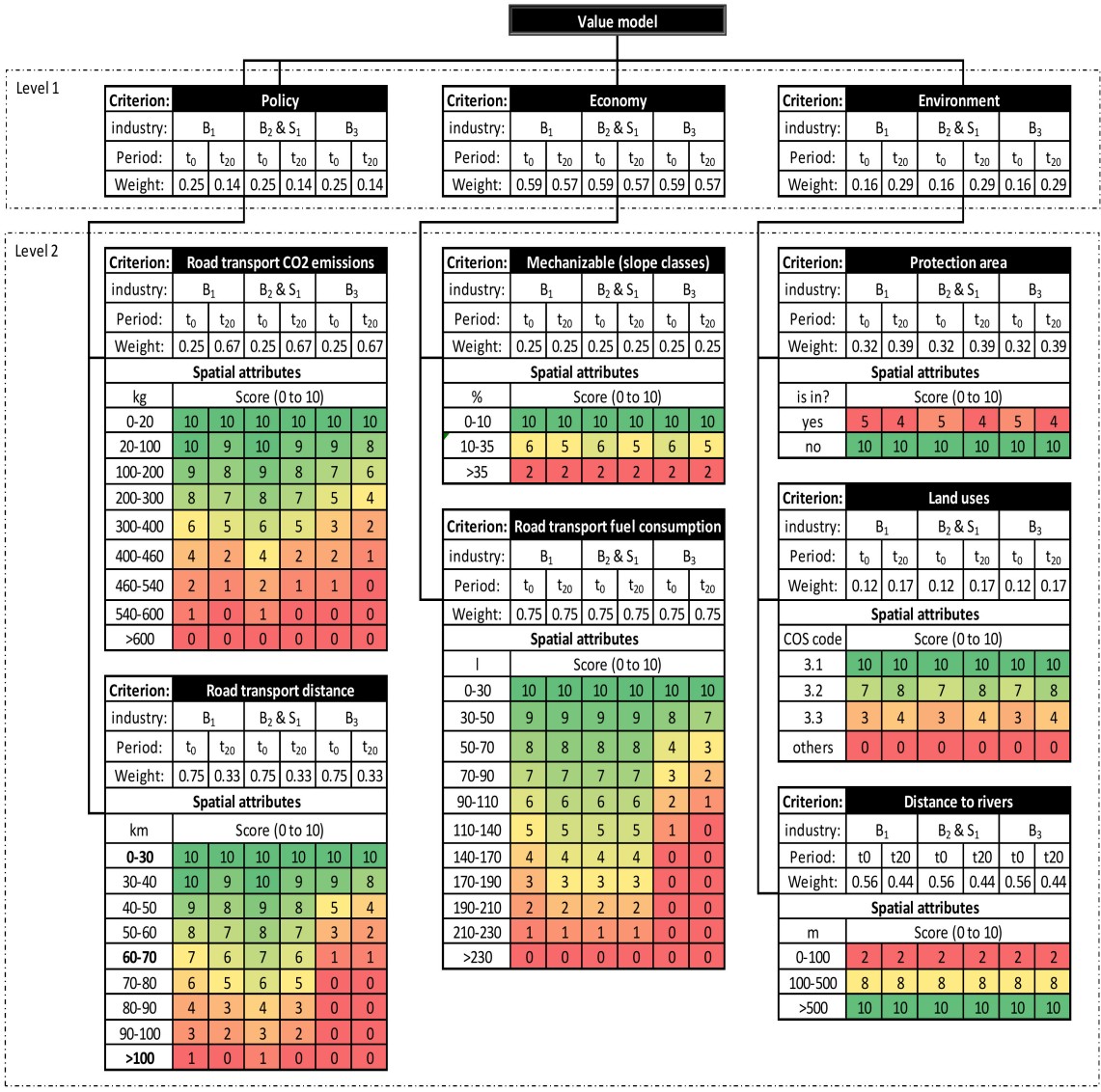

**Figure 5.** Value model for B1, B2/S1 and B3 industries in the Nordeste region, Portugal, based on AHP. Red and green represent the lowest (0) and the highest (19) score values for criteria level/alternative per industry, respectively. Other colors in the red-green gradient represent intermediate values.

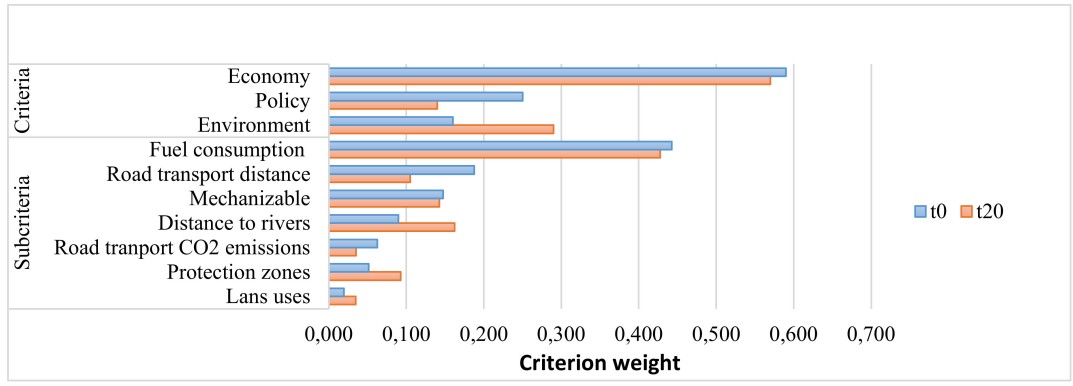

**Figure 6.** Ranking of weights for criteria and sub-criteria for $t_0$ (blue bars) used in the generic value model and trends over a period of 20 years ($t_{20}$, orange bars) in the Nordeste region, Portugal.

The price model in this study was directed exclusively to forest biomass. Although there are good examples of biomass pricing models in the literature (e.g., [58–60]), the fact that they consider costs associated to the entire supply chain made us create an utility model to associate price with the model value based on an estimate of the average price that the buyer is willing to pay associated to the maximum resource value and the average price that the seller is willing to sell for, within the boundaries of resource availability. The price model was evaluated regarding Objective 4 with the establishment of the two indicators used to prioritize scenarios:

**Id 5**: Average price of thinning (€/m$^3$)—average price of thinning resulting from the application of the value and price models for the study area;

**Id 6**: Average price of felling (€/m$^3$)—average price of felling resulting from the application of the value and price models for the study area.

### 2.5. Supply–Demand Interaction: Development of Competition Scenarios

There is often more than one industry operating simultaneously in a given area. When a particular land unit $U_{ij}$ is harvested by one industry it will become unavailable to others. In our model we used optimization (Integer Linear Programming) to define which competitor harvests which land unit. Since we worked with 1-ha cells ($U_{ij}$) that cannot be subdivided, variables are of type integer. Considering that finding a solution for problems of this type is mathematically and computationally demanding, we used a Branch and Bound (B&B) algorithm [61] to obtain an approximately optimal solution, within a defined threshold. We addressed the biomass supply–demand relationship based on the following steps:

(1) **Maximizing value**: maximization of the value (maximum resource suitability) for each spatial unit; it was assumed that all industries seek the highest resource value and are willing to pay the price corresponding to its value. Due to the large number of spatial units in the study area, we established an automatic routine in C# to formalize the problem following the Integer Linear Programming (ILP) scheme in Equation (1)

$$Max\left(\sum(f_{value})\right) \tag{1}$$

subject to

$$\forall_{T_i}\sum(quantity \cdot Element(i,j)) \in T_i \leq Max(Vol_{T_i}) \cap \geq Min(Vol_{T_i}) \tag{2}$$

$$\forall Element(i,j) \subset (0,1) \tag{3}$$

Counters

$$Sup = \sum Element(i,j) \tag{4}$$

$$Vol = \sum (quantity \cdot Element(i, j)) \tag{5}$$

Variable type

$$Int \forall Element(i, j) \in T_i \tag{6}$$

(2)  **Solving** the problem: solving the ILP scheme in (1) using an approximation of optimal solution by the B&B method applied with the open source tool lp_solve 5.5 IDE [62]; B&B was restricted to a threshold (GAP) of 10% and stopped when the first solution was found;

(3)  **Writing the solution**: production of a new forest spatial distribution considering changes in each $U_{ij}$ in time when a solution in (2) was found.

We applied the following rules to the supply–demand interaction model:

(i)  The value of the resource is established based on its demand through a set of criteria representing the valuation of demand according to each industry;

(ii)  The price of the resource is related to its value (maximum value, maximum price; minimum value, minimum price;

(iii)  Forests can be harvested when three conditions are simultaneously met:

   a.  Availability—the resource is available when it can be exploited according to economic, social and environmental conditions and local national legal frameworks;

   b.  Price—the price of the resource satisfies the two parts involved in the process: the buyer and the seller;

   c.  Quantity—there is enough resource for the sustainable operation of an industry;

(iv)  Forest resources can only be extracted (through thinning or harvesting) by one of the competing industries. When a land unit meets all the rules above, it will be used (harvested or thinned) by the industry that pays the highest price.

To evaluate the supply–demand interaction regarding objective 5 we used the following indicator:

**Id 7**: Cumulative managed area (ha)—cumulative forest area where thinning was applied over time.

Moreover, the interaction between supply and demand among industries can increase competition, which should be minimized according to objective 6, assessed based on indicator Id 8:

**Id 8**: Volume not harvested due to competition ($m^3$)—calculated as the summation of the differences between the industry harvests and the minimum value between the maximum harvest volume goal and the potential volume available for harvesting for each industry.

Finally, to assess the last objective (objective 7) we used the following two indicators:

**Id 9**: Total burned area (ha)—summation of all areas burned in the entire simulation period;

**Id 10**: Total burned volume ($m^3$)—summation of all volume lost due to fire in the entire simulation period.

## 3. Results

### 3.1. Initialization Model

According to the protocol followed in this study (Figure 3), the first simulation of the model for 10 industry scenarios over a period of 20 years considering forest biomass initialization based on a stochastic distribution (Appendix A) and broad initial volume limits (maximum goals) within each industry (Appendix C) and a set of objectives (Table 1), produced useful information regarding the dimensioning, sustainability and viability of tested scenarios. Since the possible dimensioning of B2, B3 and S1 units were unknown in advance, we stablished maximum volume goals to understand the effects of activity of these industries on supply and demand (objectives 1 and 2) which allowed the definition of informed minimum and maximum goals in a second simulation. Model assessment of the

first simulation based on the variation in standing wood in the region (Id 1) indicated that scenarios 2 and 3 led to overexploitation (Figure 7, Table 4), i.e., harvested a volume of wood that exceeded growth, therefore reducing standing volume in the region, which was considered unsustainable. In the remaining scenarios, there is an increment in standing volume in the area over time. Id 2 showed always positive volume variations in the last 5 years of simulations.

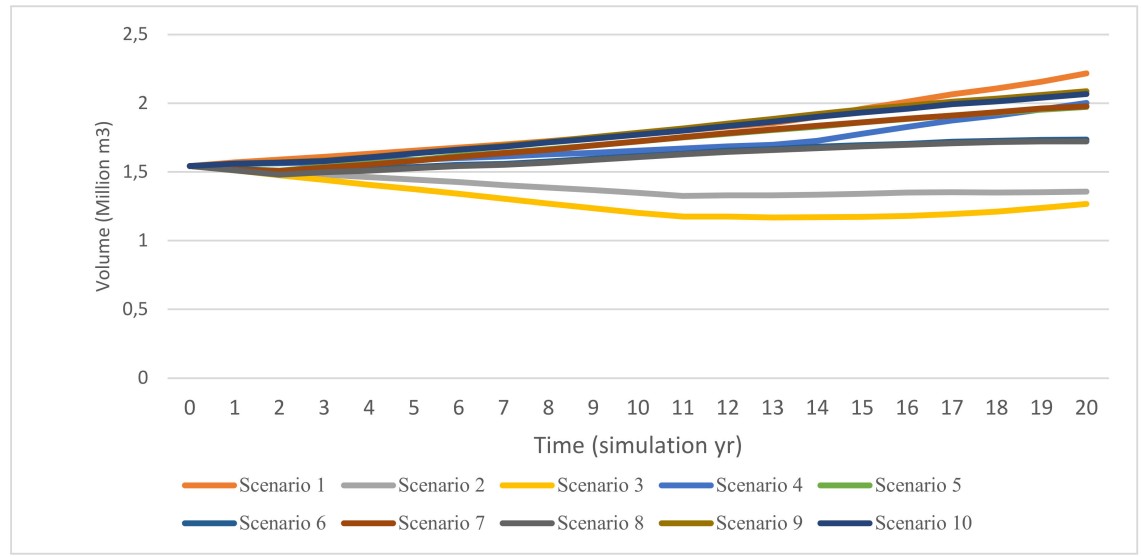

**Figure 7.** Standing volume (m$^3$) in the study area over the 20-yr simulation period in the first simulation for all scenarios considering broad volume goals (Table A4, Appendix C).

**Table 4.** Forest biomass assessment indicators outputs for the first simulation using initial volume goals in each industry/scenario (Table A4, Appendix C).

| Scenario | Obj. 1 | | Obj. 2 | Obj. 3 | Obj. 4 | | Obj. 5 | Obj. 6 | Obj. 7 | |
| | Id 1 (m$^3$/yr) | Id 2 (m$^3$/yr) | Id 3 (Y/N) | Id 4 (m$^3$) | Id 5 (€/m$^3$) | Id 6 (€/m$^3$) | Id 7 (ha) | Id 8 (m$^3$) | Id 9 (ha) | Id 10 (m$^3$) |
|---|---|---|---|---|---|---|---|---|---|---|
| 1 | 32,170 | 50,485 | Yes | 730,686 | 8.42 | 16.40 | 2647 | 0 | 5040 | 358,792 |
| 2 | −9635 | 1157 | Yes | 1,548,762 | 9.81 | 16.87 | 1815 | 497 | 4921 | 261,518 |
| 3 | −16,587 | 22,114 | Yes | 1,604,465 | 9.02 | 17.42 | 1650 | 91,071 | 4976 | 246,817 |
| 4 | 21,355 | 43,034 | Yes | 1,412,117 | 8.26 | 17.16 | 2765 | 11,029 | 5130 | 346,312 |
| 5 | 24,792 | 22,173 | Yes | 1,000,087 | 10.41 | 30.82 | 4194 | 151,260 | 5136 | 354,205 |
| 6 | 12,973 | 7729 | Yes | 1,208,860 | 10.41 | 21.80 | 2685 | 166,175 | 5068 | 324,538 |
| 7 | 25,408 | 22,980 | Yes | 1,011,611 | 10.34 | 30.75 | 4338 | 225,035 | 4996 | 338,116 |
| 8 | 13,018 | 6495 | Yes | 1,227,226 | 9.99 | 21.25 | 2811 | 251,117 | 5024 | 316,945 |
| 9 | 29,147 | 26,521 | Yes | 873,077 | 8.42 | 23.78 | 2476 | 2382 | 5063 | 365,358 |
| 10 | 27,857 | 27,002 | Yes | 898,211 | 10.28 | 23.70 | 2959 | 25,730 | 4957 | 357,653 |

The results of this first simulation with respect to demand showed that, in many scenarios, the maximum harvest goals established for the industries involved have been too optimistic and that the simulations allowed initial harvesting corresponding to the maximum volume goals, then decreasing when demand is reduced by operations and competition (see Figure 8 for scenario 3, Table 4 and Supplementary Materials for the remaining scenarios). Scenarios 2 and 3 presented the highest demand of wood (Id 4), near 1.5 and 1.6 million m$^3$, respectively. For the supply–demand interaction indicators (Id 7, Id 9 and Id 10), these same scenarios showed the lowest thinned area and the lowest area/volume affected by forest fires. Wood lost due to competition among the industry is the highest in scenarios 7 and 8 (Id 8).

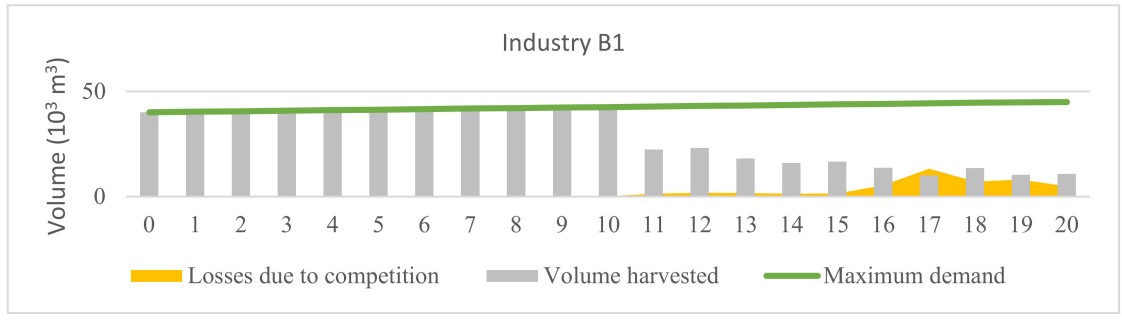

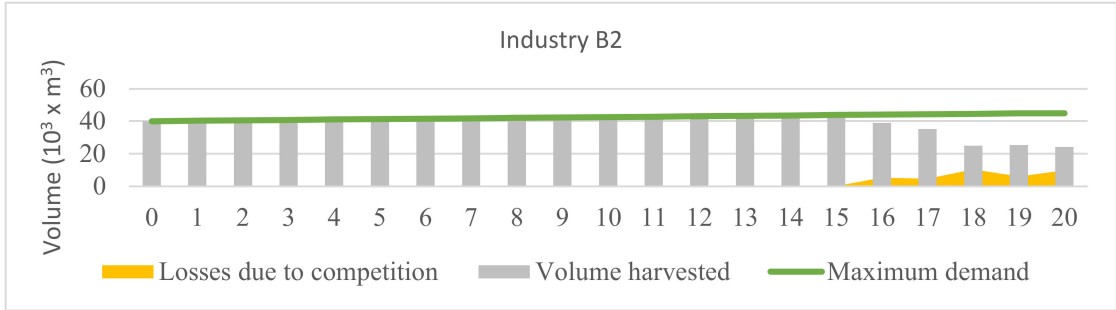

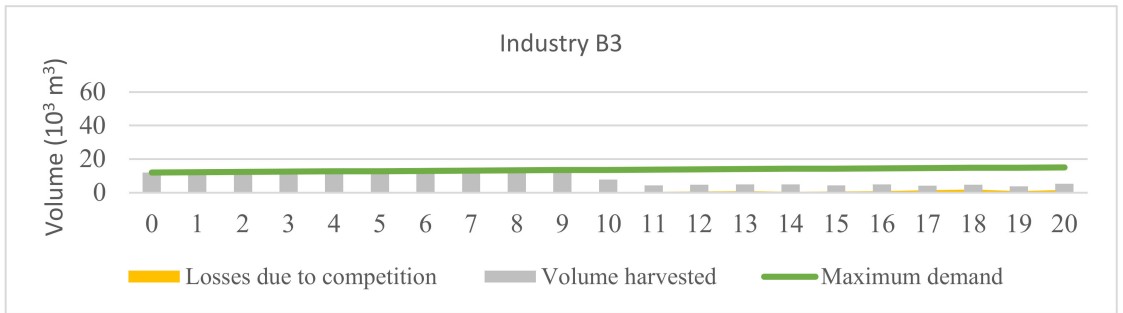

**Figure 8.** Results of simulation for Scenario 3 in terms of volume extracted by each of the industrial activities comprised (**B1**, **B2**, **B3**) with indication of maximum volume goals and volume lost due to competition among industries. The charts show that demand decreases after simulation year 10 (9 for B3) indicating that the maximum volume goal was too optimistic and must be adjusted in the next step of the protocol.

*3.2. Goals Adjustment*

The second simulation, performed with adjusted volume goals for the industries included in all scenarios (Table A5, Appendix C) (step 3 in Figure 3) and run with the forest stochastic distributions $FSD_1$ to $FSD_5$ (Appendix A) to look for the possible effects of resource distribution on the results, led to a more restricted selection of viable scenarios (Table 5, Figure 9). Scenarios 3 to 7 were not able to fulfil demand throughout the duration of the simulation period (Table 5, Figure 10). In these scenarios, simulations broke before $t_{20}$ due to the lack of biomass available for industry operation (Figure 10). Scenarios 1, 9 and 10 had the lowest impact on standing volume (Id 1 and Id2) in the region (Figure 9). These were also the scenarios harvesting less in the area (Id 4). Scenario 8 exhibited the highest effect of competition on volume (Id 8), 45,600 $m^3$ (Table 5).

This process conducted for the selection of scenarios 1, 2, 8, 9 and 10 that complied with objectives 1 and 2 established initially (Table 1). These selected scenarios were, therefore, the ones considered sustainable (Objective 1), that increased mobilisation (Objective 3), and that simultaneously scored highly in terms of the remaining Objectives. Scenarios 3, 4, 5, 6 and 7 presented irregular supply of

wood during the 20 years of simulations (Figure 10), and for this reason they were rejected (did not achieve Objective 2).

**Table 5.** Forest biomass assessment indicators outputs for the second simulation using adjusted volume goals (minimum and maximum) in each industry/scenario (Table A5, Appendix C). All values are averages of 5 simulations (FD1 to FD5).

| Scenario | Obj. 1 | | Obj. 2 | Obj. 3 | Obj. 4 | | Obj. 5 | Obj. 6 | Obj.7 | |
| | Id 1 (m³/y) | Id 2 (m³/y) | Id 3 (Y/N) | Id 4 (m³) | Id 5 (€/m³) | Id 6 (€/m³) | Id 7 (ha) | Id 8 (m³) | Id 9 (ha) | Id 10 (m³) |
|---|---|---|---|---|---|---|---|---|---|---|
| 1 | 45,350 | 46,470 | Yes | 524,588 | 8.39 | 16.56 | 2717 | 0 | 5056 | 396,398 |
| 2 | 10,838 | 7225 | Yes | 1,154,367 | 10.36 | 16.87 | 1879 | 632 | 4945 | 323,078 |
| 3 | 3607 | - | No | - | - | - | - | - | - | - |
| 4 | 37,135 | - | No | - | - | - | - | - | - | - |
| 5 | −3570 | - | No | - | - | - | - | - | - | - |
| 6 | −3496 | - | No | - | - | - | - | - | - | - |
| 7 | −13,024 | - | No | - | - | - | - | - | - | - |
| 8 | 6542 | 4031 | Yes | 1,308,383 | 9.97 | 21.14 | 2718 | 45,600 | 4949 | 304,874 |
| 9 | 17,544 | 19,280 | Yes | 1,088,890 | 8.40 | 23.64 | 2442 | 2232 | 5107 | 334,255 |
| 10 | 27,090 | 27,506 | Yes | 902,343 | 10.35 | 23.72 | 2826 | 4851 | 5063 | 358,871 |

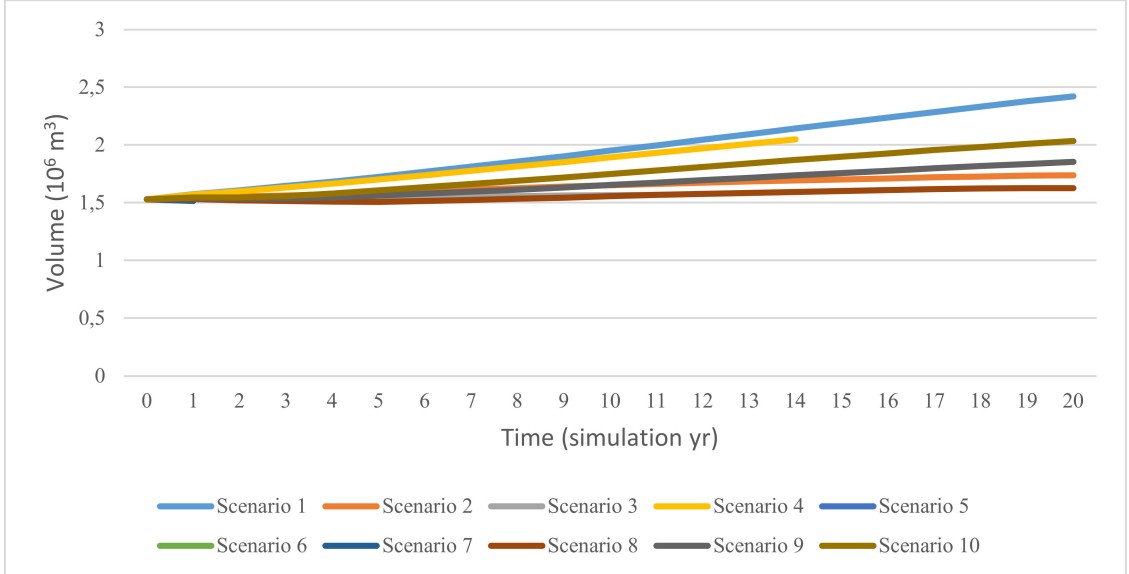

**Figure 9.** Standing volume (m³) in the study area over the 20-y simulation period in the second simulation for all scenarios considering strict minimum and maximum volume goals (Table A5, Appendix C).

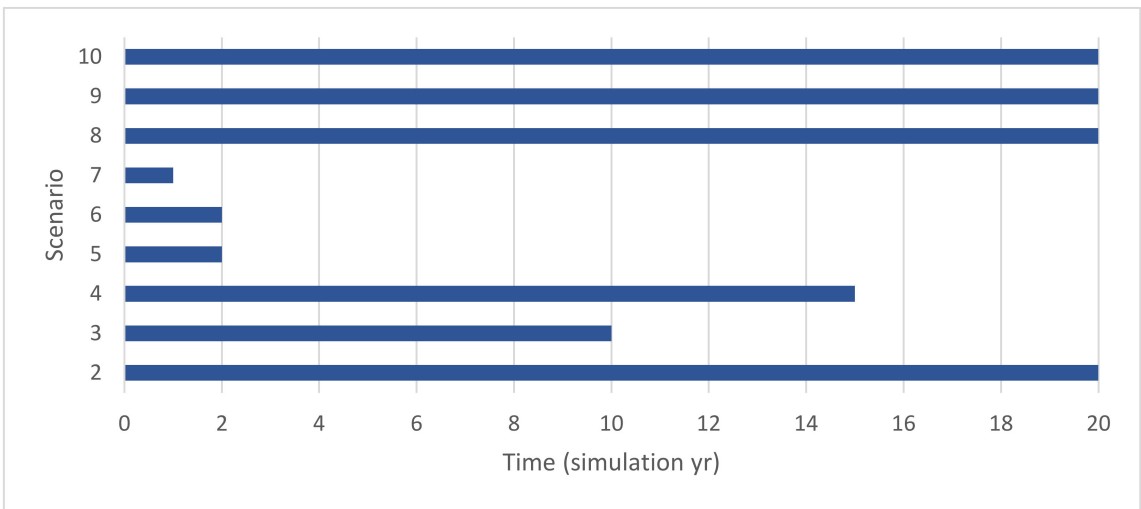

**Figure 10.** Evaluation of model performance throughout the simulation period for scenarios 2 to 10 (simulation for FD1). Bars not reaching the simulation year 20 indicate that the model broke due to lack of demand.

## 4. Discussion

### *4.1. Scenario Selection*

#### 4.1.1. The Selection Process Based on Objectives

The objectives for scenario selection proposed in this study (Table 1) were established in terms of the sustainability of forest management in the region. We wanted them mainly to increase wood mobilization while assuring that additional forest management would not be driven to overexploitation in the region and that, at the same time, harvested volume would not significantly increase competition but would increase the value and price of forest biomass, outputs that we consider to be able to motivate the forest sector in the region.

The 10 scenarios addressed initially (Table 2) provided a reasonable diversity of forest industry alternatives to be tested in the region, including new industries in Bragança and Vimioso (biomass for energy plants, sawmills) combined with the already-existing pellet production plant in Chaves (Scenario 1), which was useful for exploring a wide range of possibilities for industry investments in this region. At the end of a relatively complex process, which derives from the complexity of decision making in forestry when integrating natural dynamics of forest growth, forest management, industry operation, competition among industries, and interactions among forest biomass supply and demand, it was possible to select of a restricted number of scenarios of viable sustainable forest mobilization, whose relative and absolute impact in the region in terms of mobilization was addressed with the support of a diversity of indicators.

#### 4.1.2. Best Scenario Analysis

The final step of the protocol (Figure 3) involved the comparison of selected scenarios to look for the best industry scenario(s) for the region considering criteria such as wood mobilization in terms of volume and area, competition among industries and resource valorisation. This was done based on the objectives in Table 1 and indicators in Table 5, and, in addition, on a series of simple rates calculated as fractions of indicator values in each scenario with the same indicator in scenario 1 (second simulation), with the exception of the ratio relative to Id 8 (R 8), which was calculated as the ratio of Id 8 (volume loss due to competition) and Id 4 (volume harvested) (Table 6) due to the fact that Id 8 for scenario 1 is zero. Comparisons with scenario 1 are legitimated by the fact that this is an already existing scenario (the only among all), although it is insufficient to significantly increase forest mobilization in the region.

Comparisons with scenario 1 allow, therefore, for measuring the degree of change caused by industries relative to the current situation in terms of each of the indicators under consideration.

**Table 6.** Ratios between indicators in Table 5 relative to scenarios 2, 8, 9 and 10, and scenario 1 except for the ratio relative to Id 8 (R 8), calculated as the ratio of Id 8 (volume loss due to competition) and Id 4 (volume harvested.

|          | Obj. 1 |      | Obj. 2 | Obj. 3 | Obj. 4 |      | Obj. 5 | Obj. 6 | Obj.7 |      |
|----------|--------|------|--------|--------|--------|------|--------|--------|-------|------|
| Scenario | R 1    | R 2  | R 3    | R 4    | R 5    | R 6  | R 7    | R 8    | R 9   | R 10 |
| 2        | 0.24   | 0.16 | Yes    | 2.20   | 1.23   | 1.02 | 0.69   | 0.0005 | 0.98  | 0.82 |
| 8        | 0.14   | 0.09 | Yes    | 2.49   | 1.19   | 1.28 | 1.00   | 0.0349 | 0.98  | 0.77 |
| 9        | 0.39   | 0.41 | Yes    | 2.08   | 1.00   | 1.43 | 0.90   | 0.0020 | 1.01  | 0.84 |
| 10       | 0.60   | 0.59 | Yes    | 1.72   | 1.23   | 1.43 | 1.04   | 0.0054 | 1.00  | 0.91 |

All the selected scenarios are possible solutions to increase mobilization, each with advantages and disadvantages over the others. All affect standing volume in the region, but in all cases, the trend of standing volume in the region is positive in the entire (R 1) and in the last five years of the simulation period (R 2). When compared to scenario 1, all scenarios cause a reduction in volume growth over time which is expected due to the increasing demand and harvesting, although none of the scenarios, from the most demanding (scenario 8) to the least demanding (scenario 10), reduces available volume in the region over time. Scenario 8 is the only that includes an industry located in Vimioso (B3). However, the dimension of this industry is very small (up to 3000 m$^3$ per year). In terms of valorisation of forest biomass, most of the gain in prices was observed for felling prices in scenarios 9 and 10, which included the sawmill (S1). Here, price is 1.43 times higher than in scenario 1. Scenario 10 is the only scenario that increased area managed through thinning (accumulated managed area) in the 20 years of the simulation (R 7), slightly above the managed area in scenario 1. All other scenarios either managed the same (scenario 8) or a smaller area (scenarios 2 and 9). Although the dimension of industry in scenario 8 is small, this scenario presented the highest competition among industries (R 8). Nearly 3.5% of volume harvested corresponds to wood disputed in competition processes. Finally, there seems to be no relevant differences in burned area and volume lost due to fire in the selected scenarios in comparison to scenario 1 (R 9 and R 10). Small differences were, however, observed for volume, in particular in scenarios 2 and 9.

*4.2. Final Considerations*

4.2.1. Behaviour of the Modelling Framework

The approach we followed was able to generate near infinite combinations of variables that were not possible to ponder in this study. For example, the number and location of industries in the region, the uses of forest biomass, the variability in forest variable distribution and the possible uncertainty in environmental, social and economic aspects are factors that had to be constrained in order to formalize the problem in a viable way. Here, we tried to simplify processes and reduce uncertainty based on a value model, price and possible trends in the short term.

The heuristic framework of AppTitude® and the respective protocol applied in this research allowed the selection of viable and sustainable industrial scenario hypotheses regarding supply/demand of forest resources, location and dimensioning. The initialization process was considered satisfactory in overcoming the problem of lack of spatial data on forest stands (Appendix A). The value model resulted in a spatial distribution of a suitability index reflecting the point of view of industries over a period of 20 years (Appendix D). Similarly, the price model, that consists of a direct relationship between value and maximum and minimum prices, was revealed to be a useful approach, as indicated by the spatial distribution of prices obtained (Appendix E).

Forest management comprises a high level of uncertainty resulting from several biotic and abiotic hazards that are potentially harmful to trees, stands and the forest landscape as a whole. These hazards

are often taken into account in forest management due to their potentially high economic and ecological impact trough the integration of risk management into forest management processes and tools [63]. In the Nordeste region, the major forest hazard is fire. Wildfires occur with relatively high frequency, although fire indicators here are not as dramatic as in other regions in Portugal. Nevertheless, fire is a threat to forest products and ecosystem services [32] and burned area and fire intensity tend to increase in the future due to changes in climate and in the landscape composition and configuration. The solution found for integrating fire risk into our model as a random process under a certain probability distribution [42] is apparently a good solution, in particular if combined with an ignition probability model [43] since, on average, 98.7% of fires are human-caused [64], because this process is sensitive to social and biophysical variables and, most importantly, to dendrometric variables affected by forest management related to the several scenarios tested in this study. Other factors affecting fire ignition and behaviour, such as climate change, were not considered in this study given the short (20-y) temporal scale considered.

### 4.2.2. Limitations

The model applied in this study, as any model, assumed a considerable level of simplification of the socioecological system under consideration. Several assumptions were made for operational reasons, the first of which is the no-conflict rule in the four scenarios selected. This means that the model did not address possible conflicts between forest industry and other sectors/groups of the society with interests over forestland, assuming that thinning/felling or no thinning/felling is simply the result of supply and demand functions based on preferences of the forest industry. It did not involve, therefore, other relevant stakeholders such as conservation institutions, environmental NGOs, tourism organizations, and forest, beekeeping, or hunting and owners' associations, that have potentially conflicting interests with industry. Similarly, the model and the study did not consider the use of forests for the supply of ecosystem services that may or may not be compatible with wood-driven management. There is a growing interest in forests in the region as suppliers of highly valued ecosystem services such as carbon sequestration, mushroom production, or water regulation [65] which requires further development of the model in order to incorporate trade-offs among ecosystems services in supply–demand assessments [66], an important line for future research in the region and elsewhere. That also brings additional challenges in terms of simplification of the process considering the computational demand of such a large problem.

In terms of economic assessment, there are factors which are not addressed in this model, including competition from potential sources of demand outside the geographic boundaries of the study area and the effects of demand on the establishment of new markets and on changes in land use and land cover in the region [67]. These factors are even more complex to model due to the difficulty of establishing relationships among factors and decisions.

Lastly, this model did not attempt to find industries' best location, as in [68,69]. The objective of our study was to evaluate individual industries once their location was defined, although best location can by approximated by trial and error with the model in its present form. Future development of the model should therefore allow this possibility based on supply, demand, and supply–demand interactions in the future.

## 5. Conclusions

The FMDSS AppTitude® and the protocol applied to evaluate scenarios of increasing biomass demand were revealed to be an effective way of testing new forest industry plants in the Nordeste region (North of Portugal) in the framework of increasing forest mobilization. The solutions found (four scenarios), based in industries located in Bragança and Chaves, provide a series of good alternatives to increase wood mobilization in the region while assuring that additional forest management prevented overexploitation and that harvests did not significantly increase competition between industries but increased the value and price of forest biomass. These scenarios and the volume of wood they are able

to mobilize sustainably suggest that the forests in the region can considerably increase the sources of income and the creation of economic activity and labor not just in industry, but also in forest management and in forest logistics. This research provides reliable information that can be taken into account in forest and regional planning, which was not previously available.

## 6. Patents

AppTitude® has Intellectual Property Rights Registration no. 03/2018/262 (Spain).

**Supplementary Materials:** It is available online at http://www.mdpi.com/2076-3417/10/18/6297/s1. Includes a spreadsheet with results of all simulations, FSDs and scenarios, and indicators.

**Author Contributions:** Conceptualization, F.P.-R. and J.C.A.; methodology, F.P.-R.; software, F.P.-R.; validation, F.P.-R. and J.C.A.; writing, F.P.-R. and J.C.A.; funding acquisition, J.C.A. All authors have read and agreed to the published version of the manuscript.

**Funding:** This research was funded through the EU 7th Framework Programme for Research, Technological Development, and Demonstration (agreement no. 613762: SIMWOOD—Sustainable Innovative Mobilisation of Wood). J.C.A received partial support from Foundation for Science and Technology (FCT), Portugal through CIMO (UIDB/00690/2020). F.P_R received partial support from the Department of Economy, Industry and Competitiveness, Spanish Government, grant no. PTQ-16-08633.

**Acknowledgments:** The authors would like to acknowledge the valuable comments and recommendations made by two reviewers.

**Conflicts of Interest:** The authors declare no conflict of interest.

## Appendix A

**Table A1.** Stochastic distributions used in the initialization process: the forest stochastic distribution $FSD_0$ was used to test all the scenarios with the initialized volumes goals; FSD1 to FSD5 were used to test the top four scenarios selected according to sustainability criteria.

| | Forest Stochastic Distribution Statistics (FSD) | | | | | |
|---|---|---|---|---|---|---|
| **Indicator** | $FSD_0$ | $FSD_1$ | $FSD_2$ | $FSD_3$ | $FSD_4$ | $FSD_5$ |
| Area (ha) | 28041 | 27918 | 27971 | 27765 | 27853 | 27906 |
| Total Volume (m3) | 1.54M | 1.49M | 1.60M | 1.53M | 1.51M | 1.50M |
| AVG Age (yrs) | 17.01 | 16.5 | 17.24 | 16.86 | 16.77 | 16.95 |
| AVG Volume per land unit ($m^3$/ha) | 55.02 | 53.55 | 57.31 | 55.14 | 54.34 | 54.00 |
| AVG density (trees/ha) | 414.09 | 414.21 | 412.47 | 413.06 | 413.45 | 414.17 |
| Managed area (%) | 0 | 0 | 0 | 0 | 0 | 0 |

## Appendix B

**Table A2.** Definition of thinning and harvesting parameters per scenario and industry.

| Scenario | Type | Variable | $B_1$ | $B_2$ | $B_3$ | $S_1$ |
|---|---|---|---|---|---|---|
| 1 | Thinning | Age $_{[Min, Max]}$ (years) | [12, 30] | - | - | - |
| | | Vol. restriction ($m^3$) | ≥25 | - | - | - |
| | Final cut | Age $_{[Min, Max]}$ (years) | [30, 65] | - | - | - |
| | | Min volume ($m^3$) | ≥100 | - | - | - |
| | | $\overline{d}_{[Min, Max]}$ (cm) | [0, 65] | - | - | - |

**Table A2.** *Cont.*

| Scenario | Type | Variable | $B_1$ | $B_2$ | $B_3$ | $S_1$ |
|---|---|---|---|---|---|---|
| 2 | Thinning | Age $_{[Min, Max]}$ (years) | [12, 30] | [12, 30] | - | - |
| | | Vol. restriction (m$^3$) | ≥25 | ≥25 | - | - |
| | Final cut | Age $_{[Min, Max]}$ (years) | [30, 65] | [30, 65] | - | - |
| | | Vol. restriction (m$^3$) | ≥100 | ≥100 | - | - |
| | | $\overline{d}_{[Min, Max]}$ (cm) | [0, 65] | [0, 65] | - | - |
| 3 | Thinning | Age $_{[Min, Max]}$ (years) | [12, 30] | [12, 30] | [12, 30] | - |
| | | Vol. restriction (m$^3$) | ≥25 | ≥35 | ≥25 | - |
| | Final cut | Age $_{[Min, Max]}$ (years) | [30, 65] | [30, 65] | [30, 65] | - |
| | | Vol. restriction (m$^3$) | ≥100 | ≥100 | ≥100 | - |
| | | $\overline{d}_{[Min, Max]}$ (cm) | [0, 65] | [0, 65] | [0, 65] | - |
| 4 | Thinning | Age $_{[Min, Max]}$ (years) | [12, 30] | - | [12, 30] | - |
| | | Vol. restriction (m$^3$) | ≥25 | - | ≥25 | - |
| | Final cut | Age $_{[Min, Max]}$ (years) | [30, 65] | - | [30, 65] | - |
| | | Vol. restriction (m$^3$) | ≥100 | - | ≥100 | - |
| | | $\overline{d}_{[Min, Max]}$ (cm) | [0, 65] | - | [0, 65] | - |
| 5 | Thinning | Age $_{[Min, Max]}$ (years) | [12, 35] | [12, 35] | - | - |
| | | Vol. restriction (m$^3$) | ≥25 | ≥25 | - | - |
| | Final cut | Age $_{[Min, Max]}$ (years) | - | - | - | [35, 65] |
| | | Vol. restriction (m$^3$) | - | - | - | ≥100 |
| | | $\overline{d}_{[Min, Max]}$ (cm) | - | - | - | [30, 65] |
| 6 | Thinning | Age $_{[Min, Max]}$ (years) | [12, 30] | [12, 30] | – | - |
| | | Vol. restriction (m$^3$) | ≥25 | ≥25 | - | - |
| | Final cut | Age $_{[Min, Max]}$ (years) | [30, 65] | [30, 65] | - | [35, 65] |
| | | Vol. restriction (m$^3$) | ≥100 | ≥100 | - | ≥100 |
| | | $\overline{d}_{[Min, Max]}$ (cm) | [0, 25] | [0, 25] | - | [30, 65] |
| 7 | Thinning | Age $_{[Min, Max]}$ (years) | [12, 35] | [12, 35] | [12, 35] | - |
| | | Vol. restriction (m$^3$) | ≥25 | ≥25 | ≥25 | - |
| | Final cut | Age $_{[Min, Max]}$ (years) | - | - | - | [35, 65] |
| | | Vol. restriction (m$^3$) | - | - | - | ≥100 |
| | | $\overline{d}_{[Min, Max]}$ (cm) | - | - | - | [30, 65] |
| 8 | Thinning | Age $_{[Min, Max]}$ (years) | [12, 30] | [12, 30] | [12, 30] | - |
| | | Vol. restriction (m$^3$) | ≥25 | ≥25 | ≥25 | - |
| | Final cut | Age $_{[Min, Max]}$ (years) | [30, 65] | [30, 65] | [30, 65] | [35, 65] |
| | | Vol. restriction (m$^3$) | ≥100 | ≥100 | ≥100 | ≥100 |
| | | $\overline{d}_{[Min, Max]}$ (cm) | [0, 25] | [0, 25] | [0, 25] | [30, 65] |
| 9 | Thinning | Age $_{[Min, Max]}$ (years) | [12, 30] | - | - | - |
| | | Vol. restriction (m$^3$) | ≥25 | - | - | - |
| | Final cut | Age $_{[Min, Max]}$ (years) | [30, 65] | - | - | [35, 65] |
| | | Vol. restriction (m$^3$) | ≥100 | - | - | ≥100 |
| | | $\overline{d}_{[Min, Max]}$ (cm) | [0, 25] | - | - | [30, 65] |
| 10 | Thinning | Age $_{[Min, Max]}$ (years) | [12, 30] | [12, 30] | - | - |
| | | Vol. restriction (m$^3$) | ≥25 | ≥25 | - | - |
| | Final cut | Age $_{[Min, Max]}$ (years) | [30, 65] | [30, 65] | - | [35, 65] |
| | | Vol. restriction (m$^3$) | ≥100 | ≥100 | - | ≥100 |
| | | $\overline{d}_{[Min, Max]}$ (cm) | [0, 25] | [0, 25] | - | [30, 65] |

**Table A3.** Price intervals (range please define all abbreviated terms upon their first appearance in a figure or table (min, max)) for each industry, scenario and type of operation.

| Scenario | Period | Harvest Type | Price €/m³) | | | | | | | |
|---|---|---|---|---|---|---|---|---|---|---|
| | | | $B_1$ | | $B_2$ | | $B_3$ | | $S_1$ | |
| | | | min | max | min | max | min | max | min | max |
| 1 | $t_0$ | Thinning | 5.00 | 12.00 | - | - | - | - | - | - |
| | $t_{20}$ | Thinning | 6.00 | 14.00 | - | - | - | - | - | - |
| | $t_0$ | Felling | 12.00 | 20.00 | - | - | - | - | - | - |
| | $t_{20}$ | Felling | 14.00 | 22.00 | - | - | - | - | - | - |
| 2 | $t_0$ | Thinning | 5.00 | 12.00 | 5.00 | 12.00 | - | - | - | - |
| | $t_{20}$ | Thinning | 6.00 | 14.00 | 6.00 | 14.00 | - | - | - | - |
| | $t_0$ | Felling | 12.00 | 20.00 | 12.00 | 20.00 | - | - | - | - |
| | $t_{20}$ | Felling | 14.00 | 22.00 | 14.00 | 22.00 | - | - | - | - |
| 3 | $t_0$ | Thinning | 5.00 | 12.00 | 5.00 | 12.00 | 5.00 | 12.00 | - | - |
| | $t_{20}$ | Thinning | 6.00 | 14.00 | 6.00 | 14.00 | 6.00 | 14.00 | - | - |
| | $t_0$ | Felling | 12.00 | 20.00 | 12.00 | 20.00 | 12.00 | 20.00 | - | - |
| | $t_{20}$ | Felling | 14.00 | 22.00 | 14.00 | 22.00 | 14.00 | 22.00 | - | - |
| 4 | $t_0$ | Thinning | 5.00 | 12.00 | - | - | 5.00 | 12.00 | - | - |
| | $t_{20}$ | Thinning | 6.00 | 14.00 | - | - | 6.00 | 14.00 | - | - |
| | $t_0$ | Felling | 12.00 | 20.00 | - | - | 12.00 | 20.00 | - | - |
| | $t_{20}$ | Felling | 14.00 | 22.00 | - | - | 14.00 | 22.00 | - | - |
| 5 | $t_0$ | Thinning | 5.00 | 12.00 | 5.00 | 12.00 | - | - | - | - |
| | $t_{20}$ | Thinning | 6.00 | 14.00 | 6.00 | 14.00 | - | - | - | - |
| | $t_0$ | Felling | - | - | - | - | - | - | 25.00 | 35.00 |
| | $t_{20}$ | Felling | - | - | - | - | - | - | 27.00 | 40.00 |
| 6 | $t_0$ | Thinning | 5.00 | 12.00 | 5.00 | 12.00 | - | - | - | - |
| | $t_{20}$ | Thinning | 6.00 | 14.00 | 6.00 | 14.00 | - | - | - | - |
| | $t_0$ | Felling | 12.00 | 20.00 | 12.00 | 20.00 | - | - | 25.00 | 35.00 |
| | $t_{20}$ | Felling | 14.00 | 22.00 | 14.00 | 22.00 | - | - | 27.00 | 40.00 |
| 7 | $t_0$ | Thinning | 5.00 | 12.00 | 5.00 | 12.00 | 5.00 | 12.00 | - | - |
| | $t_{20}$ | Thinning | 6.00 | 14.00 | 6.00 | 14.00 | 6.00 | 14.00 | - | - |
| | $t_0$ | Felling | - | - | - | - | - | - | 25.00 | 35.00 |
| | $t_{20}$ | Felling | - | - | - | - | - | - | 27.00 | 40.00 |
| 8 | $t_0$ | Thinning | 5.00 | 12.00 | 5.00 | 12.00 | 5.00 | 12.00 | - | - |
| | $t_{20}$ | Thinning | 6.00 | 14.00 | 6.00 | 14.00 | 6.00 | 14.00 | - | - |
| | $t_0$ | Felling | 12.00 | 20.00 | 12.00 | 20.00 | 12.00 | 20.00 | 25.00 | 35.00 |
| | $t_{20}$ | Felling | 14.00 | 22.00 | 14.00 | 22.00 | 14.00 | 22.00 | 27.00 | 40.00 |
| 9 | $t_0$ | Thinning | 5.00 | 12.00 | - | - | - | - | - | - |
| | $t_{20}$ | Thinning | 6.00 | 14.00 | - | - | - | - | - | - |
| | $t_0$ | Felling | 12.00 | 20.00 | - | - | - | - | 25.00 | 35.00 |
| | $t_{20}$ | Felling | 14.00 | 22.00 | - | - | - | - | 27.00 | 40.00 |
| 10 | $t_0$ | Thinning | 5.00 | 12.00 | 5.00 | 12.00 | - | - | - | - |
| | $t_{20}$ | Thinning | 6.00 | 14.00 | 6.00 | 14.00 | - | - | - | - |
| | $t_0$ | Felling | 12.00 | 20.00 | 12.00 | 20.00 | - | - | 25.00 | 35.00 |
| | $t_{20}$ | Felling | 14.00 | 22.00 | 14.00 | 22.00 | - | - | 27.00 | 40.00 |

## Appendix C

Broad initial and adjusted volume limits (maximum, and minimum and maximum goals) for industries in scenarios under consideration.

**Table A4.** Initialization maximum volume goals per scenario and industry for used in the first simulation (FSD0).

| Scenario | Period | \multicolumn Volume Goals (m$^3$) Per Industry | | | | | | | |
|---|---|---|---|---|---|---|---|---|---|
| | | B1 | | B2 | | B3 | | S1 | |
| | | min | max | min | max | min | max | min | max |
| 1 | $t_0$ | - | 40,000 | - | - | - | - | - | - |
| | $t_{20}$ | - | 50,000 | - | - | - | - | - | - |
| 2 | $t_0$ | - | 40,000 | - | 40,000 | - | - | - | - |
| | $t_{20}$ | - | 45,000 | - | 45,000 | - | - | - | - |
| 3 | $t_0$ | - | 40,000 | - | 40,000 | - | 12,000 | - | - |
| | $t_{20}$ | - | 45,000 | - | 45,000 | - | 15,000 | - | - |
| 4 | $t_0$ | - | 40,000 | - | - | - | 12,000 | - | - |
| | $t_{20}$ | - | 45,000 | - | - | - | 15,000 | - | - |
| 5 | $t_0$ | - | 20,000 | - | 30,000 | - | - | - | 30,000 |
| | $t_{20}$ | - | 30,000 | - | 40,000 | - | - | - | 40,000 |
| 6 | $t_0$ | - | 20,000 | - | 30,000 | - | - | - | 30,000 |
| | $t_{20}$ | - | 30,000 | - | 40,000 | - | - | - | 40,000 |
| 7 | $t_0$ | - | 20,000 | - | 30,000 | - | 12,000 | - | 30,000 |
| | $t_{20}$ | - | 30,000 | - | 40,000 | - | 15,000 | - | 40,000 |
| 8 | $t_0$ | - | 20,000 | - | 30,000 | - | 12,000 | - | 30,000 |
| | $t_{20}$ | - | 30,000 | - | 40,000 | - | 15,000 | - | 40,000 |
| 9 | $t_0$ | - | 20,000 | - | - | - | - | - | 30,000 |
| | $t_{20}$ | - | 20,000 | - | - | - | - | - | 30,000 |
| 10 | $t_0$ | - | 20,000 | - | 5000 | - | - | - | 30,000 |
| | $t_{20}$ | - | 20,000 | - | 5000 | - | - | - | 30,000 |

**Table A5.** Minimum and maximum adjusted volume goals per scenario and industry. Scenarios are those selected as viable in the previous step.

| Scenario | Period | \multicolumn Volume Goals (m$^3$) Per Industry | | | | | | | |
|---|---|---|---|---|---|---|---|---|---|
| | | B1 | | B2 | | B3 | | S1 | |
| | | Min | Max | Min | Max | Min | Max | Min | Max |
| 1 | $t_0$ | 10,000 | 25,000 | - | - | - | - | - | - |
| | $t_{20}$ | 10,000 | 25,000 | - | - | - | - | - | - |
| 2 | $t_0$ | 10,000 | 20000 | 10,000 | 30,000 | - | - | - | - |
| | $t_{20}$ | 10,000 | 25,000 | 10,000 | 35,000 | - | - | - | - |
| 3 | $t_0$ | 10,000 | 20,000 | 15000 | 30,000 | 5000 | 10,000 | - | - |
| | $t_{20}$ | 10,000 | 20,000 | 15000 | 35,000 | 5000 | 12500 | - | - |
| 4 | $t_0$ | 10,000 | 20,000 | - | - | 5000 | 10,000 | - | - |
| | $t_{20}$ | 10,000 | 20,000 | - | - | 5000 | 12,500 | - | - |
| 5 | $t_0$ | 10,000 | 20,000 | 10,000 | 20,000 | - | - | 10,000 | 30,000 |
| | $t_{20}$ | 10,000 | 20,000 | 10,000 | 20,000 | - | - | 10,000 | 30,000 |
| 6 | $t_0$ | 10,000 | 20,000 | 10,000 | 20,000 | - | - | 10,000 | 30,000 |
| | $t_{20}$ | 10,000 | 20,000 | 10,000 | 20,000 | - | - | 10,000 | 30,000 |
| 7 | $t_0$ | 10,000 | 20,000 | 10,000 | 20,000 | 5000 | 10,000 | 10,000 | 30,000 |
| | $t_{20}$ | 10,000 | 25,000 | 10,000 | 20,000 | 5000 | 10,000 | 10,000 | 30,000 |
| 8 | $t_0$ | 2000 | 10,000 | 5000 | 15,000 | 1000 | 3000 | 25,000 | 40,000 |
| | $t_{20}$ | 2000 | 10,000 | 5000 | 15,000 | 1000 | 3000 | 25,000 | 40,000 |
| 9 | $t_0$ | 0 | 20,000 | - | - | - | - | 25,000 | 40,000 |
| | $t_{20}$ | 0 | 20,000 | - | - | - | - | 25,000 | 40,000 |
| 10 | $t_0$ | 0 | 20,000 | 0 | 5000 | - | - | 15,000 | 30,000 |
| | $t_{20}$ | 0 | 20,000 | 0 | 5000 | - | - | 15,000 | 30,000 |

## Appendix D

Spatial distribution of forest biomass demand based of the value model and its dynamics from $t_1$ to $t_{20}$ in the region for tested industrial plants.

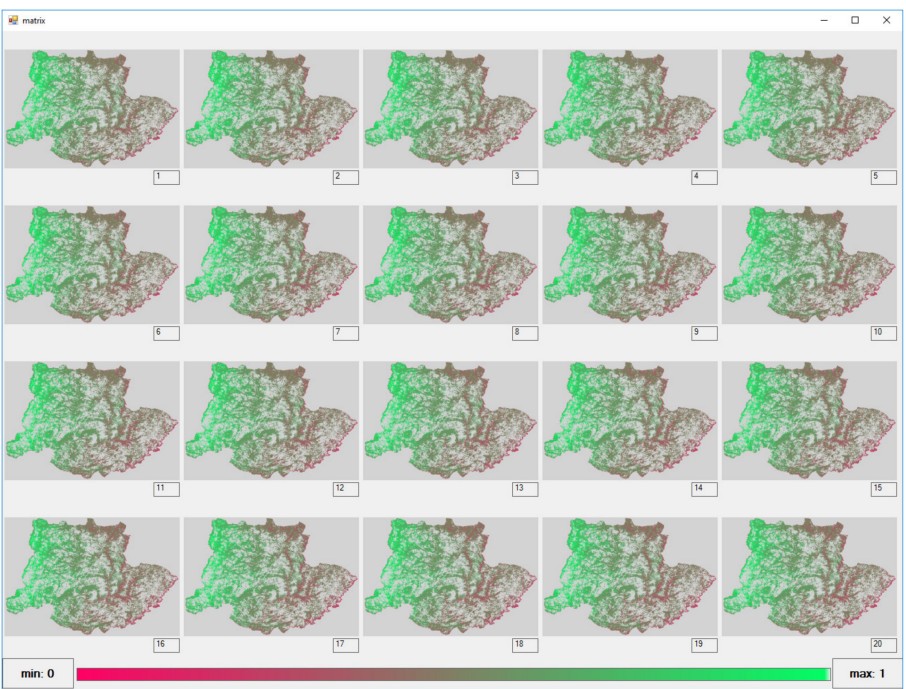

**Figure A1.** Value distribution (on a 0 to 1 scale) for Biomass I (B$_1$) over a period of 20 years (t$_1$ to t$_{20}$) in the Nordeste region, Portugal. Light green cells indicate higher suitability and light red cells lower suitability of forest units for industry.

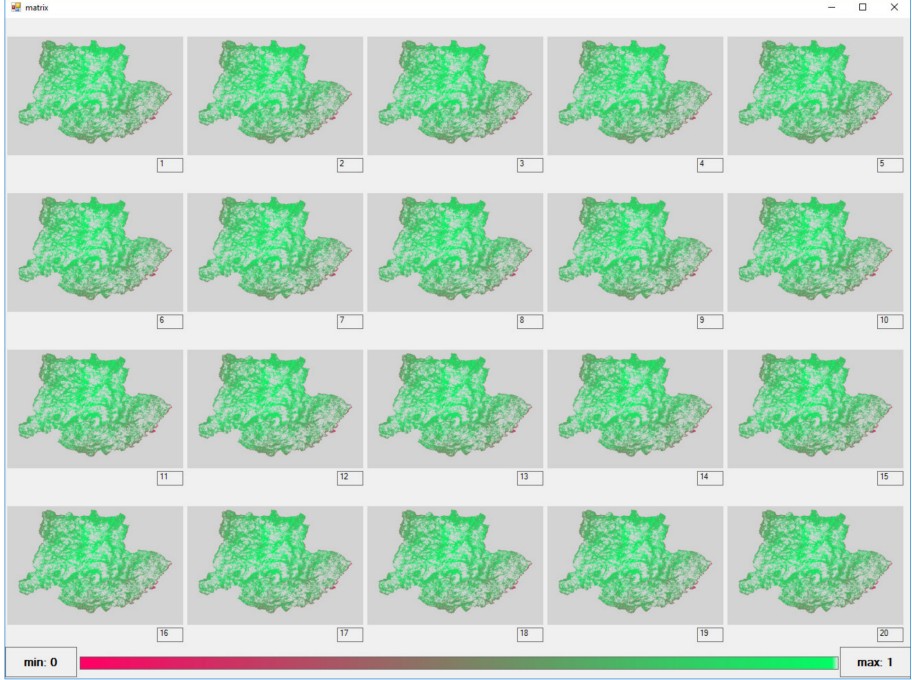

**Figure A2.** Value distribution (on a 0 to 1 scale) for Biomass II (B$_2$) and Sawmill I (S$_1$) over a period of 20 years (t$_1$ to t$_{20}$) in the Nordeste region, Portugal. Light green cells indicate higher suitability and light red cells lower suitability of forest units for industry.

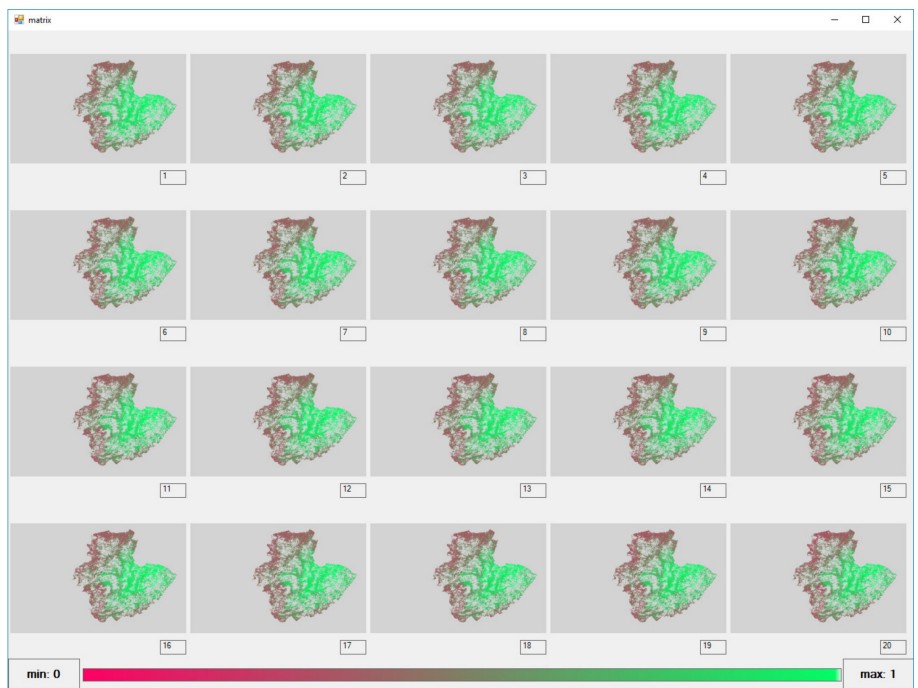

**Figure A3.** Value distribution (on a 0 to 1 scale) for Biomass III (B3) over a period of 20 years (t1 to t20. in the Nordeste region, Portugal. Light green cells indicate higher suitability and light red cells lower suitability of forest units for industry.

## Appendix E

Spatial distribution of forest biomass demand based on the price model and its dynamics from t1 to t20 in the region for tested scenarios (industrial plants).

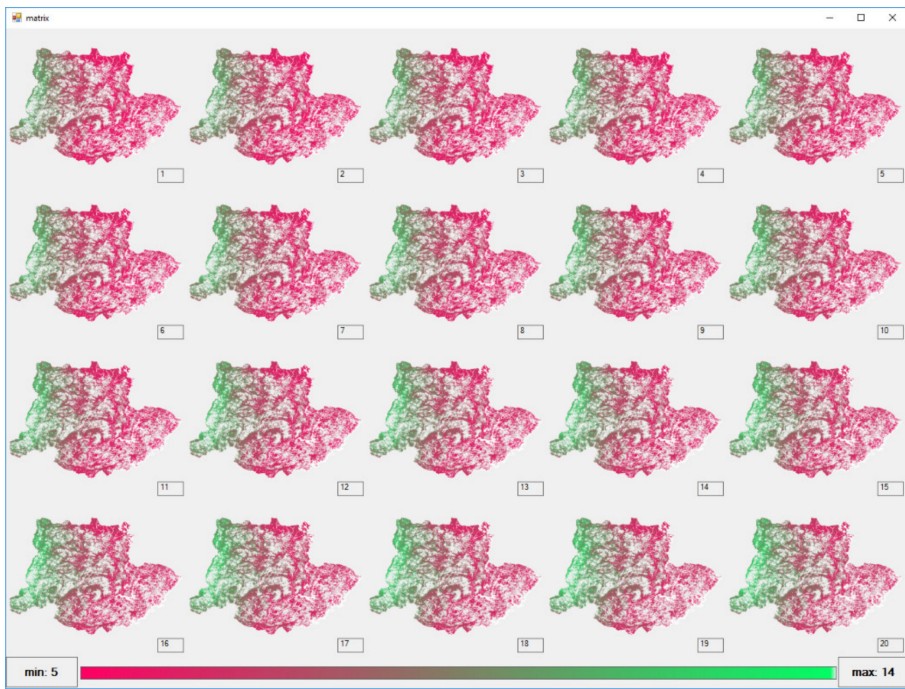

**Figure A4.** Price distribution for Scenario 1 (Biomass I (B1), thinning) over a period of 20 years ($t_1$ to $t_{20}$) in the Nordeste region, Portugal. The minimum price was 5 €/m$^3$ (light red) and the maximum 14 €/m$^3$ (light green).

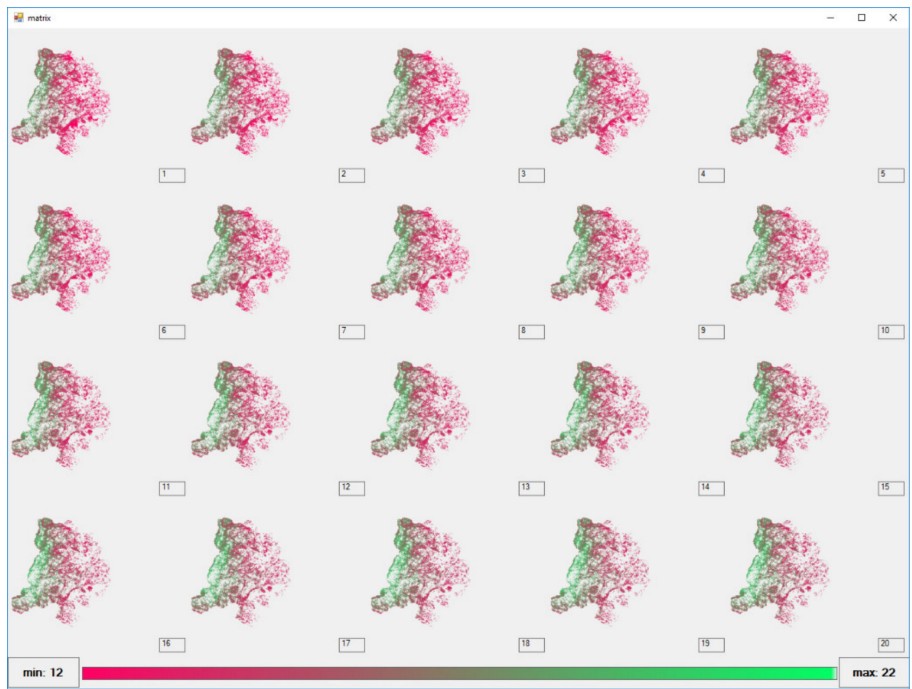

**Figure A5.** Price distribution for Scenario 1 (Biomass I (B1), felling) over a period of 20 years ($t_1$ to $t_{20}$) in the Nordeste region, Portugal. The minimum price was 12 €/m$^3$ (light red) and the maximum 22 €/m$^3$ (light green).

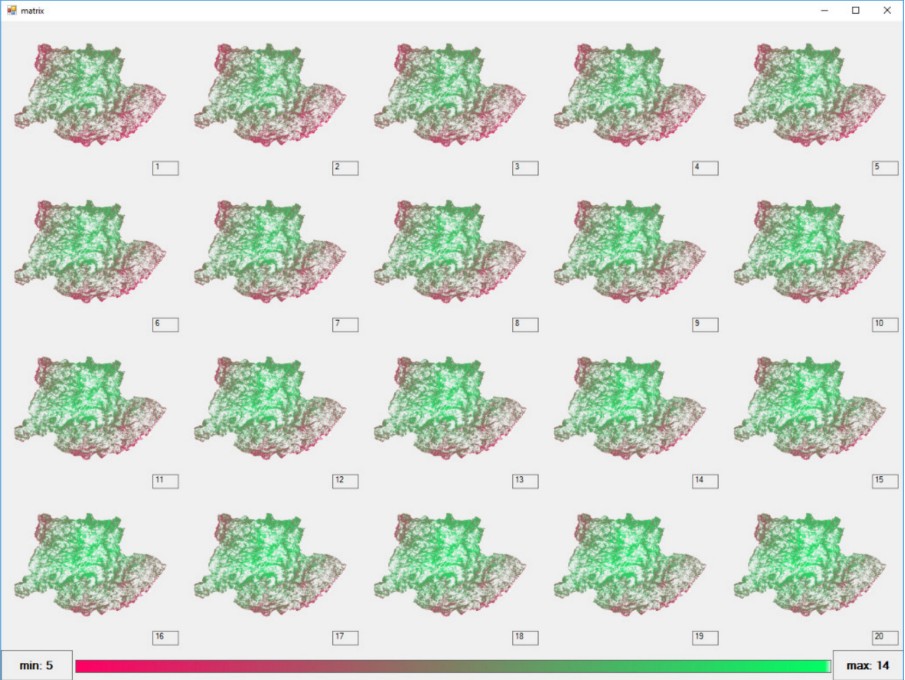

**Figure A6.** Price distribution for Scenario 1 (Biomass II (B$_2$), thinning) over a period of 20 years ($t_1$ to $t_{20}$) in the Nordeste region, Portugal. The minimum price was 5 €/m$^3$ (light red) and the maximum 14 €/m$^3$ (light green).

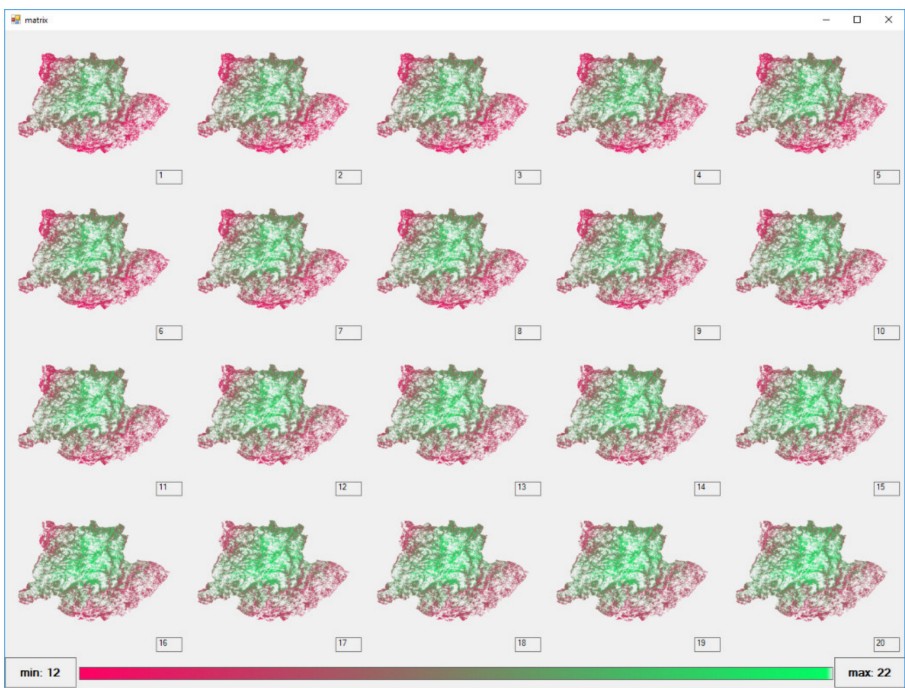

**Figure A7.** Price distribution for Scenario 1 (Biomass II (B$_2$), felling) over a period of 20 years (t$_1$ to t$_{20}$) in the Nordeste region, Portugal. The minimum price was 12 €/m$^3$ (light red) and the maximum 22 €/m$^3$ (light green).

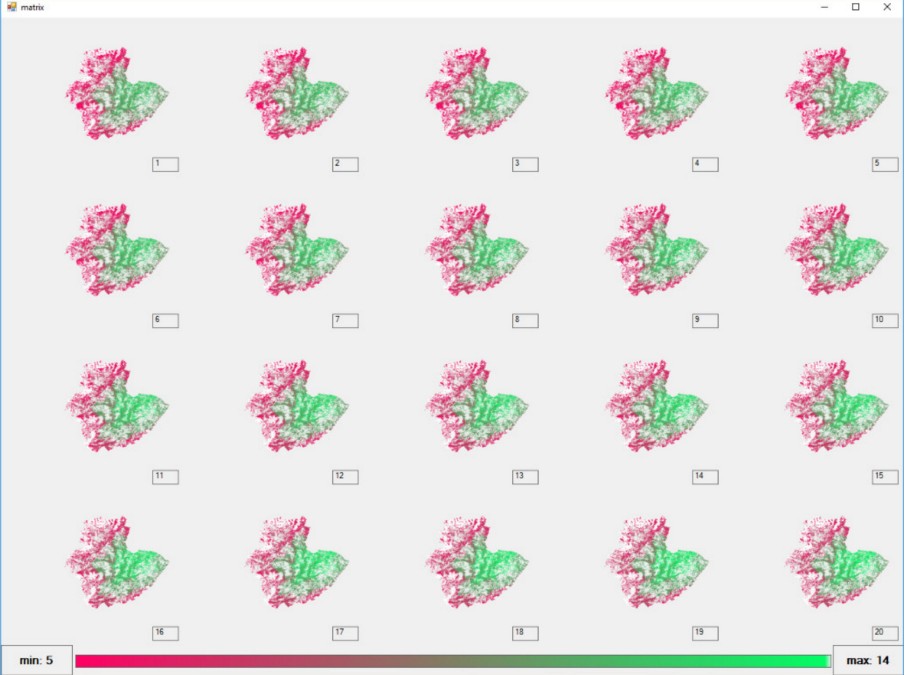

**Figure A8.** Price distribution for Scenario 1 (Biomass III (B$_3$), thinning) over a period of 20 years (t$_1$ to t$_{20}$) in the Nordeste region, Portugal. The minimum price was 5 €/m$^3$ (light red) and the maximum 14 €/m$^3$ (light green).

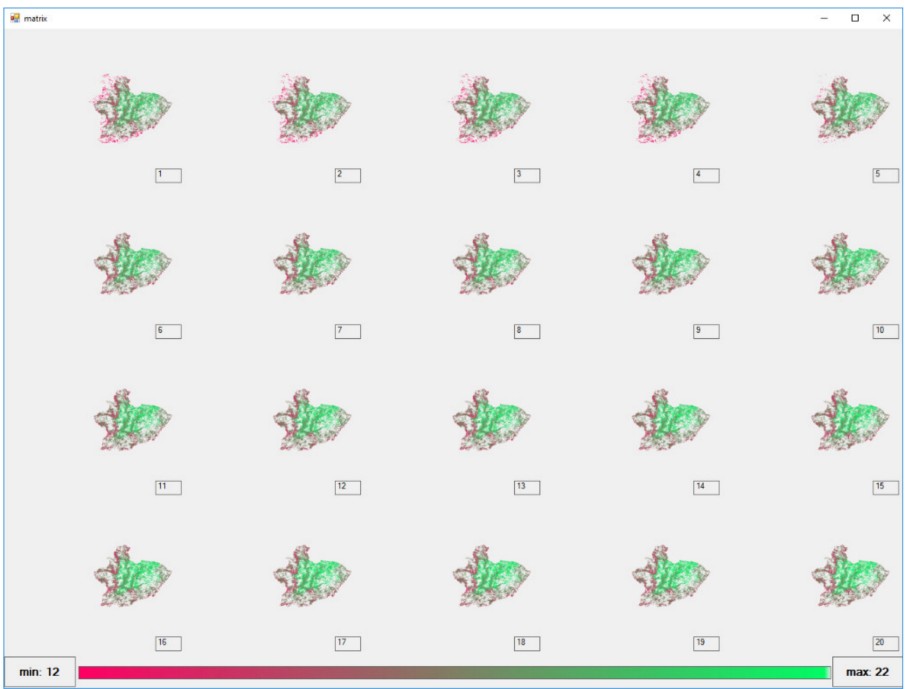

**Figure A9.** Price distribution for Scenario 1 (Biomass III ($B_3$), felling) over a period of 20 years ($t_1$ to $t_{20}$) in the Nordeste region, Portugal. The minimum price was 12 €/m$^3$ (light red) and the maximum 22 €/m$^3$ (light green).

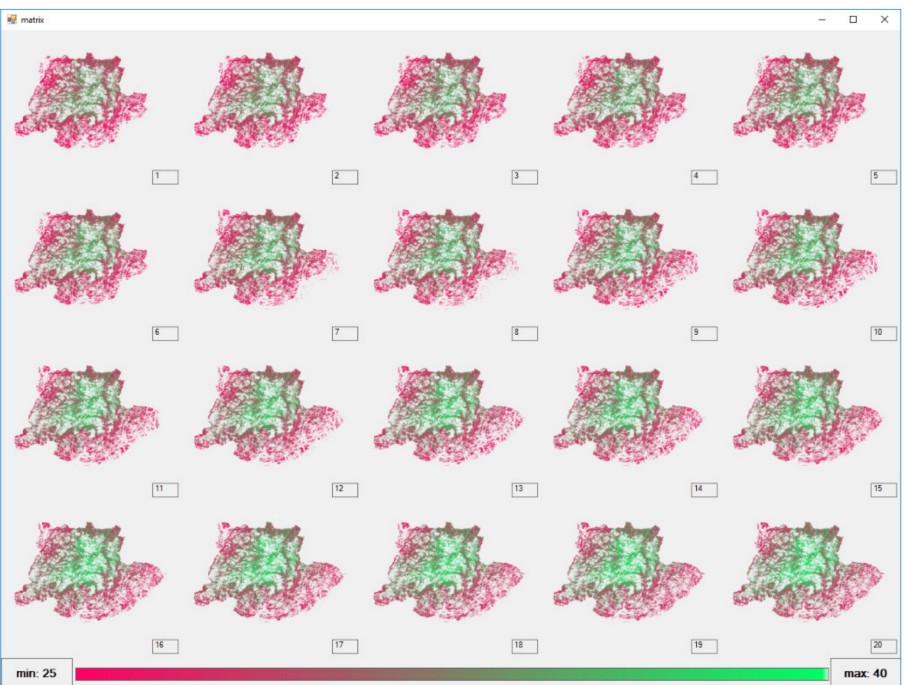

**Figure A10.** Price distribution for Scenario 1 (Biomass III ($S_1$), felling) over a period of 20 years ($t_1$ to $t_{20}$) in the Nordeste region, Portugal. The minimum price was 12 €/m$^3$ (light red) and the maximum 22 €/m$^3$ (light green).

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
