# Peer review of "Evaluation of Forest Industry Scenarios to Increase Sustainable Forest Mobilization in Regions of Low Biomass Demand"

_applsci, doi:10.3390/app10186297_

Round 1

Reviewer 1 Report

The abstract should describe how the model fulfilled the role. 

Figure 2:  The paper talks about feedback but there are few options shown in the process diagram.   

The Objects are not clearly stated for objectives 1 and 2 with three being okay. 

What are the elements and criteria on meeting sustainable resource supply?  Overall, the paper needs to a better job how the language is translated into the formulation. 

I did not see how the AHP was used - typically is used to weight goals that are used inside a model to assign weights to goal or it used post-optimizaiton to evaluate the alternatives. It is not clear to me which what was performed.  

equation 1 - how is value determined; how is the criteria weight used in the model?   

Overall, very hard to read... lines 307-320 are the most difficult to read - so much is lfeft out, please have the manuscript reviewed by more experience english language user.     

I did not see any mathematical or process model the show how goals are adjusted... what is the procedure?  

Line 418 Best scenario - are they compared using the AHP post-hoc or preoptimizaiton - does it make  a difference? 

Conclusion - 

be specific about what was accomplished - what was obtained from the results.  

Author Response

Thanks for your comments. Here, we respond to all of them one by one. These responses are corresponded by changes in the revised manuscript text file using the “track changes” tool. The line numbers indicated in the responses refer to the text when changes are not visible.

The abstract should describe how the model fulfilled the role. 

We have added a short sentence at the end of the abstract (“AppTitude® revealed to be a powerful and reliable tool to assist forest planning”) to refer to this aspect of the model (lines 24-25). As a result of this addition, we made changes in the abstract to comply with the 200 words limit.

Figure 2:  The paper talks about feedback but there are few options shown in the process diagram. 

We don’t talk about feedback in the paper. The word is not used in the manuscript. However, there are feedback-like processes included in the framework and in Figure 2, like the value model tree (Expert opinion point of diagram of Figure 2) and the Conflict rules and Supply-Demand Rules. The overall process of initialization and adjustment of goals is in general a feedback loop. We didn’t make changes in the MS to address this general comment.

The Objects are not clearly stated for objectives 1 and 2 with three being okay. 

We guess that Reviewer 1 refers to objectives and not objects. We rewrote objectives i and ii to make them more clear (lines 81-83).

What are the elements and criteria on meeting sustainable resource supply?  Overall, the paper needs to a better job how the language is translated into the formulation.

Due the large scale of the assessment conducted in this MS, the criteria used on sustainable resource supply was addressed directly as part of objective 1 in Table 1 (“Assure resilience of the region to absorb the operations of all industries”) which was evaluated in simulations with indicators 1 and 2 (Lines 186-191). This aspect is central to the methodology applied and included in components of the model other than supply, namely the supply-demand interaction model where harvesting depends, among other conditions, of the presence of enough resource for the sustainable operation by an industry. The elements and criteria of sustainable supply are therefore addressed in several sections of the manuscript and we consider that these issues are clear in the MS. We have actually spent a great deal of time and energy to make sure this central aspect of the paper was presented clearly.  

I did not see how the AHP was used - typically is used to weight goals that are used inside a model to assign weights to goal or it used post-optimizaiton to evaluate the alternatives. It is not clear to me which what was performed. 

The AHP process has been used to obtain weights of criteria in the value model tree based on expert opinion (Lines 248-250: “AHP was used in our framework to obtain weights for criteria in a decision tree …”). In Figure 5, the results of application of AHP are presented for Level 1 components (Policy, Economy and Environment) and Level 2 components for each criterion. These are given for t0 (opinion at the present) and t20 (opinion on how it could be 20 years ahead). We included AHP in the caption of Figure 5 but did not add any more methodological information on the use of AHP since we believe the comment of Reviewer 1 was more about what AHP was used for than how it was used, and the answer for that question is clearly presented in the manuscript. The application of AHP is not described in detail in the manuscript due to its already long length.  

equation 1 - how is value determined; how is the criteria weight used in the model?

The value in equation 1 is determined solving the value model tree that resulted from the relative value of each criterion of level 2 (Figure 5) weighted by each attribute score. The results of all the procedures are the maps of value for each industry (because there are criteria that depends on location): Figure D1, D2 and D3. Eq. 1 is applied to establish the hierarchy of forest units where each individual industry wants to extract wood from (felling or thinning) that combined with price and quantity models defines wood resources harvested in each year of simulation.  We considered the value model sufficiently explained in 2.4.2. (Value and price models). To introduce more detail about these processes would involve expanding the size of the manuscript that is already long. For those reasons, and since the answers to the questions of Reviewer 1 are already presented in the manuscript, we decided not to add more information to the MS.

Overall, very hard to read... lines 307-320 are the most difficult to read - so much is lfeft out, please have the manuscript reviewed by more experience english language user.     

These particular lines (307 to 320) refer to a small section of the methods relative to rules of the supply-demand interaction model. We have rewritten these rules (lines 309-316) to improve the level of English and to make them easier to read and understand.

I did not see any mathematical or process model the show how goals are adjusted... what is the procedure? 

Adjustment of goals is an essential process in the methodology applied. This procedure is explained in detail and discussed throughout the MS in several sections, in particular section 2.2. (Approach and model development), 3.1. (Initialization model) and 3.2. (Goals adjustment), among several other sections and annexes. This adjustment is done after the initialization procedure. All the processes were run in AppTitude®, a software and DSS tool that integrates a series of procedures that are applied mathematically according the flow diagram shown in Figure 2. It would not be possible to include all mathematical processes in the paper due to the extension of equations necessary to include. These processes are presented conceptually in the diagram of simulations (Figure 2) but not mathematically for that reason.  

Line 418 Best scenario - are they compared using the AHP post-hoc or preoptimizaiton - does it make  a difference? 

In this point (4.1.2 Best scenario analysis) the best scenarios are compared directly based on indicators and rates calculated among scenarios. This means that the indicators are compared quantitatively or mathematically, without incorporation of decision opinion. However, we discussed that the best scenario depends at least on the opinion or expectations of investors. There is not exactly the best scenario but there are three possible best scenarios that meet all the goals. This is discussed in 4.1.2 (Best scenario analysis). Therefore, no AHP was used in this analysis of best scenarios. Since the question is very general, we didn’t make any change in the manuscript.

Conclusion - 

be specific about what was accomplished - what was obtained from the results.  

We consider that the conclusion is already very specific. The conclusion is short and objective indicating what the results of the study imply in terms of the objectives of the work as well as the impact of the selected scenarios on forest mobilization. We can’t think of a more specific way of presenting the conclusion.  

Reviewer 2 Report

The manuscript is well organised and well prepared. The only, minor issue I find here is the length of the paper. There are lots of details of the research presented and not sure if all necessary, although extensive literature review is apreciated. However, I would suggest the editor to make a final call, since I believe the research deserves to be published in present form.

Author Response

The manuscript is well organised and well prepared. The only, minor issue I find here is the length of the paper. There are lots of details of the research presented and not sure if all necessary, although extensive literature review is apreciated. However, I would suggest the editor to make a final call, since I believe the research deserves to be published in present form.

We agree with reviewer 2: the paper is long. However, this version results already of months of work to reduce the size of the MS from its initial length. Considering the complexity of the process and the number of components, steps, criteria, indicators, etc., involved in the work, it was not possible for us to make it shorter. Eventually, Annexes A to E can be removed from the manuscript reducing the total length of the MS in 9 pages, although these annexes are important for the understanding of the work. We, therefore, leave that decision to the Editor of the SI.